# Preference Optimization for Reasoning with Pseudo Feedback

**Fangkai Jiao**[1,3†]   **Geyang Guo**[4†]   **Xingxing Zhang**[2]   **Nancy F. Chen**[3,1]   **Shafiq Joty**[5,1]

**Furu Wei**[2]

[1]Nanyang Technological University   [2]Microsoft Research   [3]I²R, A*STAR

[4]Georgia Institute of Technology   [5]Salesforce Research

## Abstract

Preference optimization techniques, such as Direct Preference Optimization (DPO), are frequently employed to enhance the reasoning capabilities of large language models (LLMs) in domains like mathematical reasoning and coding, typically following supervised fine-tuning. These methods rely on high-quality labels for reasoning tasks to generate preference pairs; however, the availability of reasoning datasets with human-verified labels is limited. In this study, we introduce a novel approach to generate pseudo feedback for reasoning tasks by framing the labeling of solutions to reason problems as an evaluation against associated *test cases*. We explore two forms of pseudo feedback based on test cases: one generated by frontier LLMs and the other by extending self-consistency to multi-test-case. We conduct experiments on both mathematical reasoning and coding tasks using pseudo feedback for preference optimization, and observe improvements across both tasks. Specifically, using Mathstral-7B as our base model, we improve MATH results from 58.3 to 68.6, surpassing both NuminaMath-72B and `GPT-4-Turbo-1106-preview`. In GSM8K and College Math, our scores increase from 85.6 to 90.3 and from 34.3 to 42.3, respectively. Building on Deepseek-coder-7B-v1.5, we achieve a score of 24.3 on LiveCodeBench (from 21.1), surpassing `Claude-3-Haiku`. [1]

## 1 Introduction

Large language models (LLMs) have demonstrated exceptional capabilities in reasoning tasks such math reasoning and coding (Roziere et al., 2023; Dubey et al., 2024; Guo et al., 2024). A *de facto* pipeline for enhancing the reasoning capabilities of LLMs involves further exposing them to reasoning specific data through continued pre-training or supervised fine-tuning (Roziere et al., 2023; Dubey et al., 2024; Yu et al., 2023; Tang et al., 2024; Dong et al., 2023), followed by preference learning techniques such as direct preference optimization (DPO; Rafailov et al. (2023)) or proximal policy optimization (PPO; Schulman et al. (2017)). Both DPO and PPO depend on reliable labels for reasoning problems to generate preference pairs and train reward models (Lightman et al., 2024; Uesato et al., 2022). Unfortunately, reasoning datasets with large-scale, human-verified labels remain limited, and scaling them through domain experts is becoming increasingly time-consuming and expensive, particularly as LLMs continue to evolve in capabilities (Burns et al., 2024; Bowman et al., 2022), which greatly limits the potential of preference learning methods such DPO and PPO.

Scalable oversight (Bowman et al., 2022) demonstrates that the annotation effort of human experts can be significantly reduced with the assistance of non-expert LLMs. However, complete elimination of human annotation remains unattainable. Building on this, Khan et al. (2024a) further reduced labeling costs by incorporating a debating mechanism, though this approach is constrained to reason-

---

[†]Work done during internship at Microsoft Research.

[1]The code is released at: https://github.com/microsoft/unilm/tree/master/PFPO

ing tasks with a finite answer space (e.g., multiple-choice questions). Other works have employed self-consistency-based answers or their variants as pseudo-labels to filter self-generated solutions (Huang et al., 2022; Yang et al., 2024c), but these methods struggle to generalize to reasoning tasks that lack explicit answer labels (e.g., coding).

To address these challenges, we frame the labeling of solutions to reasoning problems as the evaluation of these solutions against the *test cases* of the problems. For tasks with explicit answer labels (e.g., mathematical reasoning and multiple-choice questions), we treat them as cases with a single test pair, where the input is empty, and the output is the answer label. In contrast, for tasks without explicit answer labels (e.g., coding), we consider them as problems with multiple test case pairs. A solution to a reasoning problem is deemed correct if and only if it passes all associated test cases. Sample solutions generated by an LLM for the same problem can be validated using the test case suite, with correct and incorrect solutions used to construct preference pairs for DPO training or to train a reward model for PPO. In this paper, we propose two types of pseudo feedback (i.e., pseudo test cases) for reasoning problems, both of which eliminate the need for human experts and can be applied at scale. First, we explore pseudo feedback from frontier LLMs, where we decompose the process of creating pseudo test cases into multiple steps to ensure that each step is manageable for frontier LLMs. Intuitively, if an LLM can pass test cases carefully curated by a stronger LLM, it is likely to provide a correct solution. Previous work Wang et al. (2022); Snell et al. (2024) has demonstrated that self-consistency improves the reasoning performance of LLMs. Based on this insight, we introduce a second form of pseudo feedback, utilizing self-consistency from our policy LLM, which is of vital importance when frontier LLMs are no longer available. Unlike the method in Wang et al. (2022), which is limited to single-test-case problems, our self-consistency feedback is designed to generalize to problems with multiple test cases. We also find that these two types of pseudo feedback complement each other and can be applied iteratively in a pipeline. We conducted experiments on both mathematical reasoning and coding using pseudo feedback for preference optimization and we observe improvements across both tasks. Specifically, using Mathstral-7B as our base model, we improved our MATH results from 58.3 to 68.6, surpassing both NuminaMath-72B and `GPT-4-Turbo-1106-preview`. In GSM8K and College Math, our results increased from 85.6 to 90.3 and from 34.3 to 42.3, respectively. Building on Deepseek-coder-7B-v1.5, we achieved a score of 24.3 on LiveCodeBench (from 21.1), surpassing `Claude-3-Haiku`.

In a nutshell, our contribution in this paper can be summarized as follows:

- We formulate the labeling of solutions to reasoning problems as the process of evaluating them against the associated *test cases*, which facilitates preference optimization.

- We explore two types of pseudo feedback based on *test cases*: one created from frontier LLMs and the other derived from generalized self-consistency w.r.t. multiple test cases.

- Experiments on mathematical reasoning and coding demonstrate the superiority of these two types of feedback. We also find they can be applied in a pipeline and iteratively to further improve the reasoning performance.

## 2 RELATED WORK

LLMs exhibit remarkable capabilities by tuning on high-quality data annotated by experts or more advanced models (Achiam et al., 2023). However, these external annotations can be costly, posing a challenge to further enhance model's performance. Inspired by the natural evolution process of human intelligence, researchers explore self-evolution methods (Tao et al., 2024) that enable models to autonomously acquire, refine, and learn from their own knowledge. Some works (Wang et al., 2023b; Ding et al., 2024) reformulate the training objective to directly model performance improvement. Others tune the model with its own responses. They first filter the model's outputs relying on ground truth labels (Zelikman et al., 2022; Wang et al., 2024b), expert annotations (Dubey et al., 2024), or more advanced models (Yang et al., 2024a; Kirchner et al., 2024), and then use the resulting refined examples for supervised or contrastive learning (Chen et al., 2024b; Yuan et al., 2024). However, they still depend on external supervision and cannot extend to larger unlabeled datasets. Recent work (Huang et al., 2022) constructs pseudo labels via self-consistency, but the improvement is limited, possibly due to model collapse (Shumailov et al., 2023; Alemohammad et al., 2023).

For related work about mathematical reasoning (Wei et al., 2022; He-Yueya et al., 2023; Chen et al., 2021b; 2022; Lightman et al., 2024; Wang et al., 2024a; Jiao et al., 2024; Lai et al., 2024; Cobbe et al., 2021b; Li et al., 2022a; Weng et al., 2022; Yu et al., 2023; Luo et al., 2023; Mitra et al., 2024; Yue et al., 2023) and code generation (Guo et al., 2024; DeepSeek-AI et al., 2024; Nijkamp et al., 2023b;a; Zelikman et al., 2022; Li et al., 2023a; 2022b; Wei et al., 2024; Le et al., 2022; Liu et al., 2023; Dou et al., 2024; Weyssow et al., 2024), we will discuss them in Appendix F due to the limitation of space.

## 3 METHOD

In reasoning tasks such as mathematical reasoning and coding, the solution to a problem can be verified using a *standard* answer or a set of test cases. This property makes it possible to automatically create preference pairs for an LLM solving reasoning tasks and further improve reasoning capabilities of the LLM with preference optimization. However, annotating reasoning problems with answers or test cases manually is expensive and time consuming. As a result, this process is difficult to executed in large scale. Therefore, we propose PFPO (Pseudo-Feedback Preference Optimization), a method to automatically create *pseudo* answers or test cases to facilitate preference learning. In this section, we first introduce preference optimization for reasoning in Section 3.1 (assuming gold answers or test cases are available). Then we will go to details of PFPO, which creates pseudo answers or test cases.

### 3.1 PREFERENCE OPTIMIZATION FOR REASONING

Suppose we have a set of reasoning problems $x$ with their test cases $T$: $\mathcal{D} = \{(x_i, T_i)\}_{i=1}^{|\mathcal{D}|}$, where $T = \{\langle i_1, o_1 \rangle, \langle i_2, o_2 \rangle, \ldots, \langle i_{|T|}, o_{|T|} \rangle\}$ and $\langle i_k, o_k \rangle$ is the input-output pair of a test case. Note that $T$ is a generalized representation for either a collection of test cases or the gold answer for problem $x$. If $x$ is a coding problem, $T$ is a set of test cases to verify the correctness of the corresponding solution of $x$. While if $x$ is one of the other reasoning problems such as mathematical reasoning or multi-choice science questions, there is only one test case in $T = \{\langle i, o \rangle\}$ and the input $i$ is empty. For example, "`compute 1 + 1`" is a math question with $i = \emptyset$ as its test case input and $o = 2$ as its test case output.

Given a reasoning problem $x$ and its test cases $T$, we are ready to evaluate the correctness of a solution $y$ produced by an LLM $\pi_\theta$ as follows:

$$r = \frac{1}{|T|}(\sum_{k=1}^{|T|} \mathbb{1}(g(y, i_k) = o_k)) \tag{1}$$

where $g(\cdot, \cdot)$ is a function to either execute the solution $y$ or extract the answer from $y$. In the most strict form, $y$ is a correct solution to problem $x$ when $r = 1$. Otherwise (i.e., $r < 1$), $y$ is an incorrect solution. Note that in mathematical reasoning, there is only one test case and $r \in \{0, 1\}$.

Note that given a problem $x$ and its corresponding test cases $T$, the process of verifying an arbitrary solution $y$ does not need any human labeling effort. We can construct preference pairs for an LLM automatically as follows. First, we use an LLM $\pi_\theta$ to sample $N$ solutions $Y = \{y_1, y_2, \ldots, y_N\}$ for problem $x$ and obtain their verification results $R = \{r_1, r_2, \ldots, r_N\}$. To further improve $\pi_\theta$, we can use PPO (Schulman et al., 2017) to optimize these feedback online or use DPO (Rafailov et al., 2023) to do preference optimization offline. In this work, we employ DPO due to its simplicity. Then, we create preference pairs from $R$ and valid pairs $(y_w, y_l)$ requires $r_w = 1$ and $r_l < 1$.

$$\mathcal{P} = \{(y_w, y_l) | r_w = 1, r_l < 1, r_w \in R, r_l \in R\} \tag{2}$$

Given these valid preference pairs, We optimize our LLM $\pi_\theta$ using the following objective:

$$\mathcal{L}_{\text{DPO}}(\pi_\theta; \pi_{\text{ref}}; \mathcal{D}) = -\mathbb{E}_{x \in \mathcal{D}, y_w, y_l \sim \pi_\theta(\cdot|x)} \left[ \log \sigma \left( \beta \log \frac{\pi_\theta(y_w|x)}{\pi_{\text{ref}}(y_w|x)} - \beta \log \frac{\pi_\theta(y_l|x)}{\pi_{\text{ref}}(y_l|x)} \right) \right] \tag{3}$$

where $\pi_{\text{ref}}$ is the reference model before the DPO stage (usually it is the model of the supervised fine-tuning stage). $\beta$ is a hyper-parameter to control the distance between $\pi_\theta$ and $\pi_{\text{ref}}$.

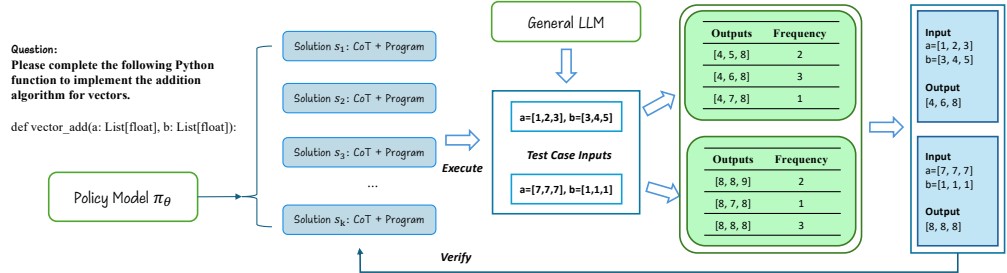

Figure 1: The process of employing self-consistency (*i.e.,* majority voting) to construct pseudo test cases for code generation problem. The outputs owing the highest frequency will be treated as pseudo outputs for verifying generated programs.

## 3.2 PSEUDO FEEDBACK PREFERENCE OPTIMIZATION FOR REASONING

In this section, we introduce how to obtain pseudo feedback for reasoning problems and the method of leverage them for preference learning.

### 3.2.1 PSEUDO FEEDBACK FROM FRONTIER LLM

**Single-Test-Case Feedback** For single-test-case reasoning tasks such as mathematical reasoning, the input is *explicitly* given in the problem itself. Therefore, we do not need to create new test cases. Given a problem $x$, we can use a frontier LLM to generate a solution $\tilde{y}$ and extract its *pseudo* answer $g(\tilde{y}, \cdot)$ as our pseudo feedback (see Equation 1). The solution $y \sim \pi_\theta(\cdot|x)$ from our model is likely to be correct if $g(y, \emptyset) = g(\tilde{y}, \emptyset)$:

$$r = \mathbb{1}(g(y, \emptyset) = g(\tilde{y}, \emptyset))) \tag{4}$$

Since solutions of many reasoning datasets used for LLM supervised fine-tuning are created by frontier LLM (Taori et al., 2023; Tang et al., 2024), we can re-use the SFT datasets and extract the pseudo feedback as a *free lunch*.

**Multi-Test-Case Feedback** For multi-test-case reasoning tasks such as coding, test cases for coding problems are usually not available and manually label them are expensive. We choose to generate pseudo test cases by prompting frontier LLMs. There are three steps to generate test cases as shown in Figure 1:

- Step 1: Given a problem $x$, generate input test cases $\mathcal{I} = \{\tilde{i}_1, \tilde{i}_2, \ldots, \tilde{i}_K\}$ by prompting a *general*[2] LLM.
- Step 2: Generate pseudo (code) solutions $\mathcal{Y} = \{y'_1, y'_2, \ldots, y'_{|\mathcal{Y}|}\}$ for problem $x$ using a frontier LLM.
- Step 3: Generate pseudo output test cases $\mathcal{O} = \{o'_1, o'_2, \ldots, o'_K\}$ using majority voting by executing all solutions in $\mathcal{Y}$ for each input test case in $\mathcal{I}$.

The output test case $o'_k$ corresponds to the input $\tilde{i}_k$ is obtained as follows: after executing all pseudo solutions, we obtain a set of candidate pseudo output $\mathcal{O}'_k = \{g(y'_1, \tilde{i}_k), g(y'_2, \tilde{i}_k), \ldots, g(y'_{|\mathcal{Y}|}, \tilde{i}_k)\}$. The output test case $o'_k$ is the most frequent element in $\mathcal{O}'_k$:

$$o'_k = \arg\max_{o \in \mathcal{O}'_k} f(o) \tag{5}$$

where $f(o) = |\{x \in \mathcal{O}'_k \mid x = o\}|$ is a frequency function that gives the number of times an element $o$ appears in $\mathcal{O}'_k$. The resulting set of pseudo test cases is $T' = \{\langle \tilde{i}_1, o'_1 \rangle, \langle \tilde{i}_2, o'_2 \rangle, \ldots, \langle \tilde{i}_K, o'_K \rangle\}$. At this point, we can verify arbitrary solution $y$ to problem $x$ as in Equation 1.

---

[2] Here we differentiate *general* LLM with the *frontier* one as generating only the inputs is much easier compared with solving the problem itself. Thus this process does not necessarily rely on SOTA LLMs.

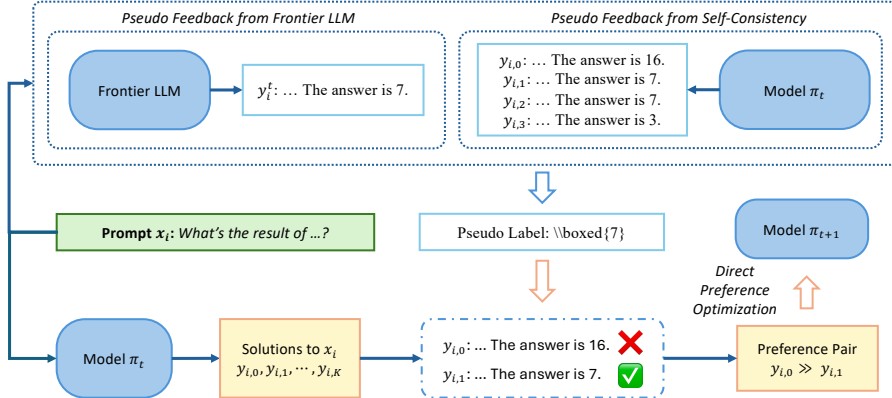

Figure 2: The overall training workflow of our method. For simplicity, we only show single step with outcome feedback for mathematical reasoning. Given an arbitrary prompt, we will sample multiple solutions from the current policy model and construct preference pairs according to pseudo feedback from frontier LLM or self-consistency. Finally, the constructed preference pair will be used to improve the policy model through DPO.

Note that we do not choose to generate both input and output test cases in a single step by prompting LLMs, as Gu et al. (2024) have pointed out that, generating the test case output based given input is a challenging task, which requires strong reasoning capabilities of LLMs. Also note that the single-test-case pseudo feedback described earlier is essentially equivalent to multi-test-case method feedback with number of test cases equals to one and the input test case is empty.

### 3.2.2 PSEUDO FEEDBACK FROM SELF-CONSISTENCY

Methods above leverage frontier LLMs to create pseudo feedback. We can alternatively create feedback from our own policy model $\pi_\theta$ to facilitate self-improvement without external guidance. We start from the method for the multi-test-case reasoning tasks, since the single-test-case counterpart is a special case of it. Specifically, we re-use the input test cases generated in Step 1 (Section 3.2.1). The main difference starts from Step 2. In the second step, we use our policy model $\pi_\theta$ to sample pseudo solutions. The pseudo output in the third step is also based on executing all pseudo solutions from our policy model $\pi_\theta$. We can apply the same process to single-test-case reasoning tasks such as mathematical reasoning, which is equivalent to using majority voted answer from $\pi_\theta$ samples as pseudo feedback. We can again use Equation 1 to verify the correctness of solutions from $\pi_\theta$.

### 3.2.3 PREFERENCE LEARNING UNDER PSEUDO FEEDBACK

Given the problem $x$ and the pseudo feedback (i.e., test cases) $T'$ we have created, the preference optimization process is as follows. We first sample $N$ solutions $Y = \{y_1, y_2, \ldots, y_N\}$ to problem $x$ from our policy $\pi_\theta$. We then obtain our verification results $R = \{r_1, r_2, \ldots, r_N\}$ using Equation 1 (i.e., executing all solutions on all test cases). We then move to create preference pairs using a different method as in Equation 2:

$$\mathcal{P}_o = \{(y_w, y_l)|r_w \geq \epsilon, r_w - r_l > \sigma, r_w \in R, r_l \in R\} \tag{6}$$

where $\epsilon$ and $\sigma$ are two hyper-parameters controlling the quality lower-bound of positive samples and the margin, respectively. Because our pseudo test cases may contains errors and if a solution $y_k$ is required to pass all test cases, we may end up with no positive solutions for problem $x$. As a result, if a solution passes enough tests ($r_w \geq \epsilon$) and significantly more tests than another solution ($r_w - r_l > \sigma$), we treat them ($(y_w, y_l)$) as an eligible preference pair.

The above preference pairs in $\mathcal{P}_o$ are based on outcome feedback of test cases. Recent studies (Wang et al., 2024a; Jiao et al., 2024) demonstrate that the outcome feedback can be used to estimate the expected returns of intermediate reasoning steps, which can help the model produce better reasoning trajectory. Motivated from this, we also construct the step-level process preference data. Following pDPO (Jiao et al., 2024), given the solution prefix $\hat{y}$, we employ the same policy model to sample $M$ completions following $\hat{y}$, and treat the averaged outcome feedback $\hat{r}$ of the completions as the

expected returns of $\hat{y}$.

$$\hat{r} = \mathbb{E}_{y \sim \pi_\theta(\cdot|x,\hat{y})} \, r(\hat{y} \circ y) \tag{7}$$

where $\circ$ is the concatenation operator. After that, the process preference data can be defined as:

$$\mathcal{P}_s = \{(\hat{y_w}, \hat{y_l}) | \hat{r_w} \geq \epsilon, \hat{r_w} - \hat{r_l} > \sigma, \hat{r_w} \in R, \hat{r_l} \in R\} \tag{8}$$

The final preference data we use is a combination of the outcome and process preference datasets $\mathcal{P} = \mathcal{P}_o \bigcup \mathcal{P}_s$. We use the DPO objective (Equation 3) to optimize the policy model $\pi_\theta$.

**Iterative Training** PFPO can be applied after the supervised fine-tuning (SFT) stage and we can train the policy model iteratively with both the feedback from frontier LLMs and from self-consistency. Imperially, we find applying LLM feedback first and then followed by self-consistency feedback rather than the opposite achieves better results. Figure 2 illustrates the process of single step.

# 4 EXPERIMENT

## 4.1 EXPERIMENTAL SETUP

**Prompt Collection** For mathematical reasoning, we followed Tang et al. (2024) to create 800K prompts under the help of GPT-4o. 500K of them are paired with one solution written by GPT-4o to construct pseudo feedback from frontier LLM. The 500K data is named as MathScale-500K, and the other prompts are called MathScale-300K. We also filtered out around 790K prompts from NuminaMath [3] by removing those that we cannot extract the predicted answers from or having appeared in the test set of MATH. For validation, we randomly sampled 2,000 question-solution pairs from the training set of MWPBench (Tang et al., 2024) after removing the questions from GSM8K (Cobbe et al., 2021a).

For code generation, we have collected the problems from the training set of APPs (Hendrycks et al., 2021a), Magicoder (Wei et al., 2024) and xCodeEval (Khan et al., 2024b), which contains 5,000, 9,000 and 6,400 questions, respectively. We remove all prompts where the test case inputs are failed to be synthesized. We randomly sampled 500 questions from the training set of APPs for validation. For APPs and xCodeEval, we use GPT-4o to generate the test case inputs. For Magicoder, we employ Mistral-Large-Instruct-2407 [4] for test case inputs generation, because of large size of the original magicoder dataset. The detailed prompt can be found in Appendix C.1.

**Evaluation** For mathematical reasoning, the performance is evaluated on the test set of MATH (Hendrycks et al., 2021b), GSM8K (Cobbe et al., 2021a), and College Math (Tang et al., 2024) by Accuracy. For code generation, we evaluate the models on HumanEval (Chen et al., 2021a), MBPP (sanitized version) (Austin et al., 2021), APPs (Hendrycks et al., 2021a), and Live-CodeBench (Jain et al., 2024) by Pass@1. Without specific clarification, all evaluations are conducted using zero-shot prompting and greedy decoding.

For simplicity, we only highlight the main results our method. For more details include the detailed source of prompts, from which checkpoints are the models initialized, please refer to Appendix A. All hyper-parameters for different experiments can be found in Appendix B.

## 4.2 EXPERIMENTAL RESULTS

### 4.2.1 MATHEMATICAL REASONING

We took two models for experiments: Llama-3.1-8B-base (Dubey et al., 2024) and Mathstral-7B-v0.1. We first conducted SFT on 500K MathScale data with GPT-4o annotation, followed by our method, with GPT-4o generated labels as pseudo feedback. As shown in Table 1, the pseudo feedback from GPT-4o can achieve consistent improvements on Llama-3.1-8b-base and Mathstral-7B. On MATH and College MATH, PFPO-LLM have made 1.2 and 4.1 averaged aboslute improvements compared with Llama-3.1-8b w/ SFT and Mathstral-7B w/ SFT, respectively.

---

[3]https://huggingface.co/datasets/AI-MO/NuminaMath-CoT
[4]https://huggingface.co/mistralai/Mistral-Large-Instruct-2407

Table 1: Overall results on mathematical reasoning benchmarks. PFPO-LLM refers to the training phase employing the pseudo feedback from frontier model (GPT-4o), while PFPO-Self indicates the phase using pseudo feedback constructed from self-generated solutions. NuminaMath-72B-CoT is built on Qwen2-72B by fine-tuning on NuminaMath. [†]: Results are from Chan et al. (2024). We employ an evaluation strategy similar to Yang et al. (2024b).

|  | MATH | GSM8K | College Math |
|---|---|---|---|
| GPT-4o-2024-0512 | 78.7 | 95.8 | 46.7 |
| GPT-4-Turbo-2024-0409 | 72.8 | 94.8 | 44.2 |
| GPT-4-Turbo-1106-preview[†] | 64.3 | — | — |
| GPT-4-0613 | 55.0 | 93.5 | 39.0 |
| NuminaMath-72B-CoT (Beeching et al., 2024) | 67.1 | 91.7 | 39.8 |
| Llama-3.1-8B-Instruct (Dubey et al., 2024) | 47.5 | 84.5 | 27.5 |
| Llama-3.1-70B-Instruct (Dubey et al., 2024) | 68.1 | 95.5 | 41.8 |
| Llama-3.1-8B-base (Dubey et al., 2024) | 20.3 (4-shot) | 56.7 (8-shot) | 20.1 (4-shot) |
| w/ SFT | 53.8 | 85.1 | 34.6 |
| w/ PFPO-LLM Iter. 0 | 55.0 | 86.6 | 35.8 |
| w/ PFPO-Self Iter. 1 | 55.9 | 87.6 | 36.6 |
| w/ PFPO-Self Iter. 2 | 56.6 | 88.9 | 37.0 |
| w/ PFPO-Self Iter. 3 | 57.0 | 88.8 | 36.7 |
| w/ PFPO-Self Iter. 4 | 57.4 | 89.1 | 37.6 |
| w/ PFPO-Self Iter. 5 | **57.8** | **89.6** | **38.0** |
| Mathstral-7B-v0.1 (Mistral AI Team, 2024b) | 58.3 | 85.6 | 34.3 |
| w/ SFT | 61.4 | 87.3 | 38.4 |
| w/ PFPO-LLM Iter. 0 | 66.7 | 90.0 | 41.3 |
| w/ PFPO-Self Iter. 1 | 67.8 | **90.8** | 42.0 |
| w/ PFPO-Self Iter. 2 | **68.6** | 90.3 | 42.2 |
| w/ PFPO-Self Iter. 3 | 68.2 | 90.4 | **42.3** |

At the second phase, we started iterative pDPO training on unseen prompts with self-consistency-based pseudo feedback. For Llama-3.1-8b, we used the prompts from NuminaMath-790K to synthesize solutions and construct pseudo feedback via self-consistency. The prompts are divided into non-overlapped splits for iterative training. As shown in the table, by employing pseudo feedback, the models achieve continuous improvements across different iterations. Specifically, our method achieves the best results at Iteration 5. Compared with Llama-3.1-8b w/ PFPO-LLM Iter. 0, it achieves consistent improvements with 2.8 on MATH, 3.0 on GSM8K, as well 2.2 on College Math, revealing the potential of iterative preference optimization via pseudo feedback from self-consistency.

For Mathstral-7B, we use the prompts in MathScale-300K for iterative training with pseudo feedback, since we did not observe improvements on NuminaMath. The prompts across different iterations are the same. As shown in the table, Mathstral-7B w/ PFPO-Self Iter. 2 achieves 1.9 absolute improvements on MATH, compared with the SFT model. And it can outperform the stronger counterparts like NuminaMath-72B-CoT[5], Llama-3.1-70B-Instruct, and GPT-4-Turbo-1106-preview, with only 7B parameters, which have demonstrated the effectiveness of pseudo feedback. Besides, we find the performance will saturate after several iterations. We discuss the possible reasons in Appendix A.2.

### 4.2.2 CODE GENERATION

For code generation, we selected Deepseek-coder-7B-v1.5-Instruct (Guo et al., 2024) for experiments. We first use GPT-4o to generate 11 program solutions for each question in the APPs training set, and use the ground-truth test cases to remove those having failed tests. The left solutions are kept to first fine-tune the base model. The resulted model is referred as *w/ SFT (APPs)*.

**Direct Preference Optimization via Test Case Execution Feedback**   As described in Section 3.2.2, we have constructed preference dataset via executing the generated programs over real or synthetic test cases. The evaluation results are shown in Table 2 and 3.

---

[5]https://huggingface.co/AI-MO/NuminaMath-72B-CoT

Table 2: Overall results (Pass@1) on program generation benchmarks. PFPO-Self refers to our training from pseudo feedback method, and the content in the brackets afterwards indicates the source of prompts. Specifically, *M.C.* refers to the prompt set of Magicoder (Wei et al., 2024), and *xCode.* is the short for xCodeEval (Khan et al., 2024b). *Introductory*, *Interview*, and *Competition* indicate the three difficulty levels of APPs. w/ (p)DPO (APPs) refers to that the execution feedback is synthesized based on the groundtruth test cases annotated in APPs training set.

| | APPs | | | | HumanEval | MBPP |
|---|---|---|---|---|---|---|
| | Overall | Introductory | Interview | Competition | | |
| GPT-4-0613 | 35.1 | 61.8 | 34.4 | 10.6 | 87.8 | 82.1 |
| GPT-4o-2024-0513 | 34.0 | 56.6 | 32.2 | 16.7 | 93.3 | 87.2 |
| Llama-3.1-8B-Instruct (Dubey et al., 2024) | 11.5 | 29.4 | 8.5 | 2.7 | 72.6 | 71.2 |
| Llama-3.1-70B-Instruct (Dubey et al., 2024) | 24.9 | 51.8 | 21.3 | 9.1 | 80.5 | 83.3 |
| Codestral-22B-V0.1 (Mistral AI Team, 2024a) | 20.3 | 45.2 | 16.9 | 5.8 | 81.1 | 78.2 |
| CodeQwen1.5-7B-chat (Qwen Team, 2024) | 8.6 | 24.1 | 16.8 | 2.0 | 85.6 | 80.5 |
| Qwen2.5-Coder-7B-Instruct (Hui et al., 2024) | 15.7 | 37.3 | 12.3 | 4.1 | 85.4 | 86.0 |
| Deepseek-coder-33B-Instruct (Guo et al., 2024) | 18.4 | 44.2 | 14.5 | 4.4 | 77.4 | 79.0 |
| Deepseek-coder-v1.5-Instruct | 14.3 | 35.7 | 10.8 | 3.2 | 75.6 | 73.9 |
| w/ SFT (APPs) | 15.4 | 37.8 | 11.6 | 4.1 | 72.0 | 72.8 |
| w/ DPO (APPs) | 16.3 | 36.2 | 13.3 | 5.3 | 74.4 | 74.3 |
| w/ pDPO (APPs) | 16.9 | 37.3 | 13.8 | 6.1 | 73.8 | 73.2 |
| w/ PFPO-LLM Iter. 0 (APPs) | 17.9 | 38.3 | 14.7 | 7.1 | 73.8 | 75.9 |
| w/ PFPO-Self Iter. 0 (APPs) | 17.4 | 37.5 | 14.8 | 5.4 | 73.2 | 75.1 |
| w/ PFPO-Self Iter. 1 (APPs & M.C.) | 18.0 | 39.2 | 14.9 | 6.2 | **79.3** | **75.5** |
| w/ PFPO-Self Iter. 2 (APPs & M.C. & xCode.) | **19.1** | **40.9** | **15.9** | **6.9** | 73.8 | 75.1 |

Table 3: Overall results on LiveCodeBench. We follow the recommended setting by sampling 10 solutions for each problem with temperature as 0.2, and estimating the Pass@1 results. The cutoff date of the test questions is from **2023-09-01** to **2024-09-01**. All results except those of our models are referenced from the official leaderboard ( https://livecodebench.github.io/).

| | Overall | Easy | Medium | Hard |
|---|---|---|---|---|
| Claude-3.5-Sonnet | 51.3 | 87.2 | 45.3 | 11.0 |
| Claude-3-Sonnet | 26.9 | 67.2 | 7.3 | 1.4 |
| Claude-3-Haiku | 24.0 | 61.3 | 5.5 | 0.9 |
| GPT-3.5-Turbo-0125 | 24.0 | 55.0 | 11.6 | 0.3 |
| Llama-3.1-70B-Instruct (Dubey et al., 2024) | 31.8 | 67.9 | 17.3 | 4.1 |
| Llama-3-70B-Instruct (Dubey et al., 2024) | 27.4 | 59.4 | 15.6 | 1.3 |
| CodeQwen1.5-7B-Chat (Qwen Team, 2024) | 16.8 | 35.9 | 10.9 | 0.3 |
| DeepSeekCoder-V2-236B (DeepSeek-AI et al., 2024) | 41.9 | 79.9 | 32.0 | 4.9 |
| Deepseek-Coder-33B-Instruct (Guo et al., 2024) | 23.4 | 56.1 | 8.6 | 0.9 |
| Deepseek-coder-7B-v1.5-Insturct | 21.1 | 51.3 | 7.4 | 0.2 |
| w/ SFT (APPs) | 22.9 | 53.0 | 10.6 | 0.2 |
| w/ DPO (APPs) | 22.9 | 53.7 | 9.4 | 1.0 |
| w/ pDPO (APPs) | 22.9 | 55.0 | 8.1 | 1.3 |
| w/ PFPO-LLM Iter. 0 (APPs) | 24.0 | 56.8 | 9.3 | **1.4** |
| w/ PFPO-Self Iter. 0 (APPs) | 23.4 | 54.2 | 10.3 | 0.7 |
| w/ PFPO-Self Iter. 1 (APPs & M.C.) | 23.7 | 55.8 | 9.5 | 1.1 |
| w/ PFPO-Self Iter. 2 (APPs & M.C. & xCode) | **24.3** | **56.8** | **9.8** | **1.6** |

First, we aim to discuss the effectiveness of fully synthetic test cases, a topic that has not yet been extensively explored. We use *w/ DPO* and *w/ pDPO* to denote methods utilizing ground truth test cases to gather execution feedback, while PFPO-Self Iter. 0 (APPs) employs the same prompt set but simulates execution feedback using pseudo test cases. From the results presented, we observe that pseudo test cases outperform ground truth ones in almost all benchmarks, with the exception of HumanEval. In

Table 4: The averaged number of test cases of each problem in the training set of APPs.

| Avg. No. | Original | Synthetic |
|---|---|---|
| Training | 5.16 | **9.95** |

particular, PFPO-Self Iter. 0 leads both 0.5 absolute improvements on APPs and LiveCodeBench compared with the groundtruth pDPO. This improvement is attributed to the potential of increasing the number of synthetic test cases, thereby reducing the false positive rate associated with missing

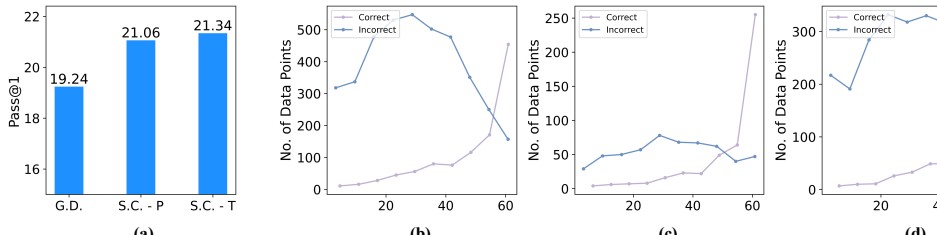

Figure 3: **(a)** The Pass@1 performance of the test set of APPs of our model variant, w/ PFPO-Self Iter. 2 (APPs & M.C. & xCode, DPO) under different strategies. *G.D.* indicates greedy decoding, *S.C.* is the short for self-consistency. We took the groundtruth test case inputs to execute the programs sampled from our policy model. For each group of test case inputs, we employ self-consistency across different programs to determine the pseudo outputs. For *S.C. - P,* we select one of the programs matching the pseudo outputs for evaluation. For *S.C. - T,* we simply check whether the pseudo outputs are consistent with the ground-truth outputs, which indeed highlights an upper bound of test case self-consistency, since sometimes we fail to select at least one solution matching all the pseudo outputs. **(b) - (d)** We measure the top-1 frequency among the outputs and average them by the number of test cases. The *x*-axis indicates the averaged top-1 frequency, and the *y*-axis demonstrates the corresponding amount of data samples. The three figures show the relationships on (b) all questions, (c) introductory-level questions, and (d) interview-level questions, respectively.

corner cases. As shown in Table 4, the questions in the APPs training set contain only 5.16 test cases on average, whereas we can ensure that each problem has approximately 10 test cases.

Besides, by comparing w/ PFPO-LLM Iter. 0 (APPs) and w/ PFPO-Self Iter. 0 (APPs), we find that the pseudo feedback generated by frontier LLM also demonstrates better results on code generation. This is also reflected through the quality of synthetic test cases, and we have a deep analysis in Appendix A.4.

**Iterative Training** Afterwards, we attempted to introduce more training data in an iterative manner to see if the pseudo feedback can continuously enhance the performance. As shown in Table 2 and 3, after three rounds of training, PFPO-Self Iter. 2 has made 3.7 and 1.4 absolute improvements on APPs and LiveCodeBench, respectively. We also observed more significant improvements on the Easy and Hard level of LiveCodeBench, *i.e.,* 3.8 and 1.4 absolute points, respectively. Besides, on HumanEval and MBPP, we did not observe extra improvements after introducing xCodeEval. This is probably because xCodeEval contains only standard input/output problems, which could lead to distribution shift. The small scale of test cases and problems in HumanEval and MBPP also make the evaluation not really stable.

## 4.3 RELATIONSHIP BETWEEN SELF-CONSISTENCY AND RELIABLE PSEUDO FEEDBACK

In this section, we will explore the impact of test-case-based self-consistency during inference for coding. We sample 64 solutions for each problem in the APPs test set and apply self-consistency over the provided test cases to select the final programs. Two strategies are developed within this process: **Self-Consistency for Programs (S.C. - P)** and **Self-Consistency for Test Cases (S.C. - T)**. Following Section 3.2.2, we execute the sampled programs **on the test case inputs annotated by the dataset** and determine the pseudo outputs for each group of inputs through majority voting.

First of all, we hope to evaluate the correctness of the self-consistency based pseudo outputs by checking the consistency with the ground truth outputs. If this is true, we treat this question is passed. This strategy is called S.C. - T. After that, we can check if there exists at lease one program among the sampled solutions such that its outputs can match all the pseudo outputs, which is called S.C. - P. In other words, for S.C. - T, we assume that there is such a program among the candidates that can obtain the expected pseudo outputs, so we only care about if the pseudo outputs are as expected. For S.C. - P, we also consider if such a program is included in the sampled solutions. To this end, S.C. - T can be considered the upper bound of self-consistency, as there are cases where no solution matches the pseudo outputs. As illustrated in Figure 3 (a), the test-case-based self-consistency also makes significant improvements during test time. Besides, by continuously sampling solutions against the pseudo outputs, the pass rate would be further improved.

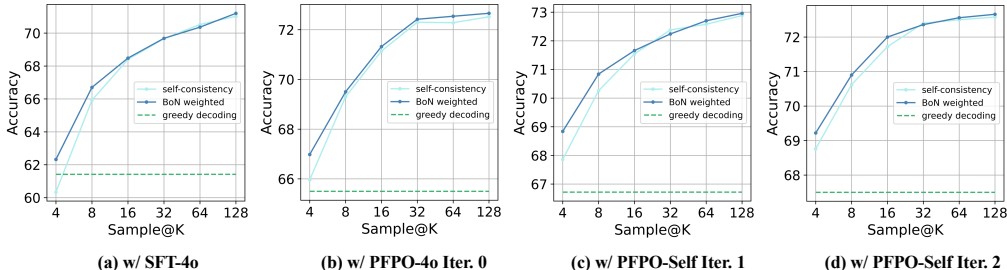

| (a) w/ SFT-4o | (b) w/ PFPO-4o Iter. 0 | (c) w/ PFPO-Self Iter. 1 | (d) w/ PFPO-Self Iter. 2 |

Figure 4: Performance comparison of four Mathstral-7B variants by scaling inference

Besides, we also want to explore the reliance of self-consistency based pseudo feedback. From Figure 3 (b) to (d), we display the distribution of top-1 output frequency averaged by the number of test cases, i.e., the confidence of the policy model, on overall, introductory-level, and interview-level problems. We find that, self-consistency over test cases provide reliable pseudo feedback when different programs achieve high consistency rate. Although the margin between positive and negative solutions is reduced with the increase of difficulty, by continuously improving the sampling budget, the uncertainty can also be reduced. This may help to both control the difficulty of training set, and lower down the possible inaccurate feedback. On mathematical reasoning, similar conclusion still holds. Figure 5 demonstrates the accumulated ratio of corrected predictions with the development of self-consistency-based prediction ratio among all candidate answers.

## 4.4 SCALING INFERENCE AND REWARD MODELING FROM PSEUDO FEEDBACK

In this section, we increase the inference budget (Snell et al., 2024) to assess how it further improves mathematical reasoning. Specifically, we explore two strategies: *self-consistency* and *best-of-N weighted* (Li et al., 2023b). The reward model used for *best-of-N weighted* is optimized on the same training set of Mathstral-7B w/ PFPO-Self Iter. 1, thus also benefiting from the pseudo feedback derived from self-consistency.

The results in Figure 4 indicate that: (i) Scaling inference makes significant improvements, and employing an extra reward model to score responses can bring more benefits, especially when using smaller sampling budget. (ii) Pseudo feedback can also enhance reward modeling. (iii) As highlighted by Snell et al. (2024), combining pseudo reward models with other inference techniques (*e.g.,* weighted best-of-N (Li et al., 2023b), lookahead search, and step-level beam search) may improve performance. We leave the relevant exploration as future work.

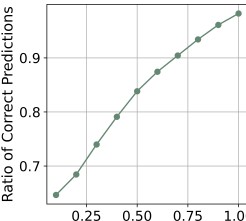

Figure 5: The accumulated ratio of correct predictions over all questions. The *x*-axis denotes the ratio of the top-1 frequency among all predicted candidates, *i.e.,*, the confidence for self-consistency. Each point in the figure indicating how many predictions are correct when the confidence is lower than the *x*-value.

## 5 CONCLUSION

In this paper, we demonstrated the potential of synthesizing pseudo-feedback from both frontier LLMs and the model itself. By incorporating feedback from GPT-4o, we improved Llama-3.1-8b-base-SFT's performance on the MATH from 53.8 to 55.0 and enhanced Mathstral-7B's performance from 61.4 to 66.7. Furthermore, by leveraging self-generated feedback based on self-consistency, we increased Llama-3.1-8b's performance to 57.8 and Mathstral-7B's to 68.6. For code generation, we introduced a novel test case-level self-consistency strategy. By employing fully self-constructed pseudo test cases, we boosted the SFT model's performance from 15.4 to 19.1 on APPs and from 22.9 to 24.3 on LiveCodeBench. Additionally, we analyzed the relationship between self-consistency and reliance on synthetic pseudo labels, offering insights for future research. For future work, we hope to combine the pseudo reward models with scaling inference techniques to seek more improvements on more challenge scenarios.

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

# A MORE RESULTS AND ANALYSIS

We list detailed results, the source of training prompts, as well as some variants not presented in the main content, in Table 9, 11, and 12.

## A.1 COMPARISON BETWEEN DPO AND pDPO

From the results across different base models and tasks, we find that pDPO usually have significantly better results than DPO. The only exception appears on HumanEval and MBPP, which is possibly due to the two benchmark does not require detailed reasoning process, and the mismatch in distribution causes performance degradation.

## A.2 VARIANTS OF ITERATIVE DPO TRAINING

In our experiments, we have employed several variants of Iterative DPO training for different considerations. For NuminaMath-790K, collecting all prompts for single iteration of DPO training can make it more challenging to avoid policy shifting, as pointed out by Xiong et al. (2024). To this end, we split the whole dataset into several parts for iterative training. During each iteration, we use around 160K prompts to collect solutions, construct pseudo feedback, and optimize the policy model.

For MathScale-300K, since the dataset is much smaller, we use all prompts across different iterations, and the process ends when we cannot observe significant improvements.

For code generation, we use a hybrid strategy. During each iteration, we introduce new dataset of prompts to be mixed with the original prompts, and collect new solutions. The pseudo test cases are also constructed based on the new solution programs. The reason is that we do not have too much data but we also hope to approach on-policy training.

During each iteration, the reference model is initialized from the policy model in last iteration, instead of the original base model.

## A.3 EFFECT FROM THE AMOUNT OF SYNTHETIC TEST CASES

As shown in Table 2 and 3, PFPO-Self achieves even higher results than the model optimized on ground-truth test cases. One possible reason behind this is we have synthesized more test cases for self-consistency to reduce the false positive rate. To further verify this point, we down-sampled the synthetic test cases to align with the number of ground truth test cases. Specifically, for each coding

problem, if the synthesized test cases exceeded the number of ground truth test cases, we randomly selected a subset of synthesized cases to match the number of ground truth test cases. Otherwise, all synthetic test cases were retained.

As shown in Table 7, the results demonstrate that reducing the number of synthetic test cases significantly decreases performance. This highlights the importance of scaling test cases for verifying program correctness effectively.

### A.4    QUALITY ANALYSIS OF SYNTHETIC TEST CASES

In this section, we will analyze the quality of the synthetic test cases. Specifically, for each question in the APPs training set, we selected a ground-truth program to evaluate the synthetic test cases. A synthetic test case is valid, if and only if given the test case input and its output matches the output of the ground-truth program. We calculate the pass rate as metric, which represents the percentage of valid test cases among the total generated. The results are shown in Table 8, where the *Model* column denotes the model optimized by the pseudo feedback constructed from the corresponding test cases.

From the table, we can conclude that: (1) The synthetic test cases constructed by the weaker policy models has already demonstrate good quality. For example, 62.68% of the test cases for first round training, i.e., PFPO-Self-Self Iter. 0 are correct. (2) Iterative training on fixed prompt set will lead to overfitting. As the saturation of the quality of synthetic test cases, the policy model will fail to collect more correct instances from the training set, and the model will thus stop learning. (3) Pseudo feedback from frontier LLMs usually demonstrates better quality. We find that the test cases synthesized from frontier LLM's solutions achieve 82.41% pass rate. Yet, we find that PFPO-LLM cannot outperform PFPO-Self Iter. 2, which is possibly due to less training data and the off-policy optimization of DPO, and can be alleviated by more rounds of traininig.

### A.5    ANALYSIS ABOUT PLATEAUED PERFORMANCE OVER ITERATIONS

As also observed by Xiong et al. (2024) and Chen et al. (2024a), iterative style DPO tends to saturate after several iterations, regardless of whether the preference pairs are human labeled or synthetic. We believe there are at least two reasons for the saturation. The first one (as mentioned in Appendix A.4) is the quality of synthetic test cases may plateau at some point. The second reason could be that as the model improves with each iteration, more problems in the training set become "too easy" for the model at its current stage. For extremely challenging problems where no correct solution can be generated despite many attempts (sampling), we are unable to create valid preference pairs and are thus unlikely to solve them in the next iteration. And the number of moderately challenging problems, which are most effective for improving the model, gradually decreases after each iteration.

One possible solution to this is introducing unseen problems in each iteration to increase the likelihood of hitting new moderately challenging problems. In the third block of Table 1, we simulated the above setting by introducing new problems across iterations. As detailed in Appendix A, we split the NuminaMath dataset into five subsets of equal size and use a different subset for each iteration. We observed consistent improvements across iterations (see w/ PFPO-Self Iter. 1 to 5 rows in Table 1). However, please note that applying the above method requires a significant amount of unseen high-quality questions in that domain. Therefore, we cannot apply this strategy with "Mathstral-7B-v0.1" based models (Mathstral-7B-v0.1 has been trained on Numina already) or in coding domain (high quality prompts in coding domain is very limited). Finally, it is worth noting that our method's advantage lies in its scalability, as no additional human annotation for answers is required when incorporating new problems.

### A.6    PERFORMANCE ON MORE CHALLENGING DATASET

We also evaluated our performance on a subset of challenging math problems to explore whether the iterative training procedure also makes continuous improvements.

For the dataset construction, we began by sampling 16 solutions generated by Mathstral-7B w/ SFT on the MATH test set. We then identified questions for which the predicted answers, even after applying majority voting, remained incorrect. This yielded a MATH-Challenging test set of 1,580

questions. As shown in Table 10, the results indicate that our method can achieve improvements even on challenging questions over iterations.

## B    Hyper-Parameters

All hyper-parameters are listed in Table 5.

## C    Prompts

### C.1    Test Case Inputs Generation

The 2-shot prompt for test case inputs generation is shown in Figure 6, 7, and 8.

### C.2    Prompt for Competition-level Code Generation with Rationale

The 1-shot prompt for competition-level code generation is shown in Figure 9, 10, and 11. During SFT, we remove the 1-shot example.

## D    More Implementation Details About pDPO

Our own implementation of pDPO include the following steps: (1) Sample multiple solutions for each problem from the policy model. (2) Supposing a non-empty line is a reasoning step, we sample multiple prefixes for each solution following a fixed ratio, where each prefix is composed of several continuous reasoning steps from the begining. If the prefix dataset is too large, we will control that each problem will have fixed amount of prefixes. In most experiments, we will set the ratio as 30% and choose the fixed amount in $\{8, 10, 20\}$, as shown in Table 5. (3) Take the prompt, problem, as well as each solution prefix to compose new input, and sample 3 completions for it. (4) Estimate the expected returns of each prefix by checking the results approached by the completions. (5) Here is the main difference with the original implementation of pDPO (Jiao et al., 2024): The authors choose to first sample small amount of prefixes as well as the estimated expected returns for training another reward model, and use the reward model to annotate the complete solutions. This is to reduce the computation resource usage. Instead, in this paper, we simply estimate expected returns for all sampled solutions within given budget, and train the policy model on the prefixes (incomplete solutions) via DPO directly.

## E    Results of Different Training Order

Table 6 compares the results of different training order. Assuming MathScale-300K and MathScale-500K share similar difficulty, the results indicate that the better quality the training data has, the earlier iteration it should be employed in.

## F    Related Work

### F.1    Mathematical Reasoning

To enhance the model's critical capability in mathematical reasoning, researchers have explored various techniques to guide the reasoning process. These efforts include prompts engineering (Wei et al., 2022), tool usage (He-Yueya et al., 2023; Chen et al., 2021b; 2022), process rewarding (Lightman et al., 2024; Wang et al., 2024a; Jiao et al., 2024; Lai et al., 2024), and verifiers (Cobbe et al., 2021b; Li et al., 2022a; Weng et al., 2022). In addition to improve frozen LLMs, researchers also work on synthesizing math related data to fine-tune them (Yu et al., 2023; Luo et al., 2023; Mitra et al., 2024; Yue et al., 2023).

## F.2 CODE GENERATION

Code generation has long been recognized as challenging due to data sparsity. The emergence of LLMs (Guo et al., 2024; DeepSeek-AI et al., 2024; Nijkamp et al., 2023b;a; Zelikman et al., 2022; Li et al., 2023a) pretrained on corpus including rich code has partially reduced this problem. Yet, it is still far from enough due to the significant gap between code completion and programming to solve complex problems. Li et al. (2022b) make earlier efforts towards competition-level code generation, which is achieved by filtering solutions from large sampling space via example test cases. Wei et al. (2024) propose to synthesize instruction tuning dataset for code generation via self-instruct (Wang et al., 2023a). Besides, Le et al. (2022); Liu et al. (2023) and Dou et al. (2024) propose to use the feedback from compiler and unit test to improve code generation through reinforcement learning. However, these approaches still rely on human annotation to obtain test cases, and thus cannot be scaled to large scale training. Dou et al. (2024) further employ the unit test based feedback for process supervision. Weyssow et al. (2024) have constructed a preference dataset for code generation annotated by GPT-3.5. Yet, the absence of test cases hinder it to be generally employed in on-policy learning.

Table 5: The hyper-parameters used in our experiments. **K - S.C.** refers to the amount of sampled solutions for each problem that are used to determine the pseudo label via self-consistency. Instead, **K - DPO** indicates the amount of sampled solutions per question used for constructing preference pairs. **No. Prefix** is the amount of sampled prefixes per question used for process DPO training. There are two kinds of values for *No. Prefix*. The integers are the exact amount, while the percentage indicates that we sampled the specific ratio of prefixes among all solutions for each problem. This is used to address the problem of limited training data. $\beta$ is the coefficient used in DPO training. $\alpha$ is the weight for NLL loss. $\epsilon$ is the reward (feedback) lower bound of positive samples to control the quality. For pDPO training, sometimes we use fraction to simply indicate the number of sampled completions as well as the successful attempts. For example, $\frac{1}{3}$ means that we sample 3 completions for each prefix, and if there is at least 1 completion is successful, it is kept. $\sigma$ is the margin to control the gap between positive and negative samples.

| Model | K - S.C. | K - DPO | No. Prefix | $\beta$ | $\alpha$ | $\epsilon$ | $\sigma$ |
|---|---|---|---|---|---|---|---|
| Mathstral-7B w/ SFT | | | | | | | |
| w/ DPO (M.S.-500k, Iter. 0) | 10 | 10 | — | 0.5 | 0.2 | 1.0 | 0.0 |
| w/ pDPO (M.S.-500k, Iter. 0) | 10 | — | 10 | 0.1 | 0.2 | $\frac{2}{3}$ | 0.0 |
| w/ pDPO (M.S.-300k-S.C., Iter. 1) | 10 | 10 | 10 | 0.5 | 1.0 | $\frac{1}{3}$ | 0.0 |
| w/ pDPO (M.S.-300k-S.C., Iter. 2) | 10 | 10 | 10 | 0.5 | 1.0 | $\frac{1}{3}$ | 0.0 |
| Llama-3.1-8b w/ SFT | | | | | | | |
| w/ DPO (M.S.-500k, Iter. 0) | 10 | 10 | — | 0.5 | 0.2 | 1.0 | 0.0 |
| w/ pDPO (M.S.-500k, Iter. 0) | 10 | 10 | 10 | 0.5 | 0.2 | $\frac{1}{3}$ | 0.0 |
| w/ pDPO (Numina-S.C. 160k, Iter. 1) | 16 | — | 8 | 0.5 | 0.2 | $\frac{1}{3}$ | 0.0 |
| w/ pDPO (Numina-S.C. 320k, Iter. 2) | 16 | 16 | 8 | 0.5 | 1.0 | $\frac{1}{3}$ | 0.0 |
| w/ pDPO (Numina-S.C. 480k, Iter. 3) | 16 | — | 8 | 0.5 | 1.0 | $\frac{1}{3}$ | 0.0 |
| w/ pDPO (Numina-S.C. 640k, Iter. 3) | 16 | — | 8 | 0.6 | 0.2 | $\frac{1}{3}$ | 0.0 |
| w/ pDPO (Numina-S.C. 790k, Iter. 3) | 16 | — | 32 | 0.6 | 0.2 | $\frac{2}{3}$ | 0.0 |
| Deepseek-coder-v1.5-chat w/ SFT | | | | | | | |
| w/ DPO (APPs) | 10 | 10 | — | 0.1 | 0.0 | 1.0 | 0.0 |
| w/ pDPO (APPs) | 10 | — | 30% | 0.1 | 0.2 | $\frac{1}{5}$ | 0.0 |
| w/ DPO (APPs - S.C.) | 10 | 10 | — | 0.1 | 0.0 | 0.5 | 0.6 |
| w/ pDPO (APPs - S.C.) | 10 | — | 30% | 0.1 | 0.0 | 0.5 | 0.4 |
| w/ DPO (APPs & M.C. - S.C.) | 10 | 10 | — | 0.4 | 0.2 | 0.5 | 0.6 |
| w/ DPO (APPs & M.C. & xCode. - S.C.) | 10 | 10 | — | 0.4 | 0.2 | 0.5 | 0.6 |
| w/ pDPO (APPs & M.C. & xCode. - S.C.) | 10 | 10 | 10 | 0.5 | 0.2 | 0.5 | 0.4 |
| w/ pDPO (APPs & M.C. - S.C.) | 10 | 10 | 20 | 0.4 | 0.2 | 0.5 | 0.4 |

Table 6: The experimental results of models trained via different orders. The prompt set for DPO training keeps the same with the original setting, where PFPO-Self employs MathScale-300K and PFPO-LLM takes MathScale-500K.

|  | MATH | GSM8K | Colledge Math |
| --- | --- | --- | --- |
| Mathstral-7B-v0.1 | 58.3 | 85.6 | 34.3 |
| w/ SFT-4o | 61.4 | 87.3 | 38.4 |
| w/ PFPO-LLM Iter. 0 | 66.7 | 90.0 | 41.3 |
| w/ PFPO-Self Iter. 1 | 67.8 | **90.8** | 42.0 |
| w/ PFPO-Self Iter. 0 | 64.6 | **89.8** | **39.4** |
| w/ PFPO-LLM Iter. 1 | **65.2** | 89.1 | 39.3 |

Table 7: The experimental results exploring the influence by the amount of synthetic test cases. *w/ down-sampled synthetic test cases* refers to the setting that we reduce the amount of synthetic test cases to which is equal and less than the that of the ground-truth test cases annotated for each problem in the training set of APPs.

|  | APPs | LiveCodeBench |
| --- | --- | --- |
| DeepSeek-coder-v1.5-chat | 14.3 | 21.1 |
| w/ SFT | 15.4 | 22.9 |
| w/ golden test cases | 16.9 | 22.9 |
| w/ synthetic test cases | **17.4** | **23.4** |
| w/ down-sampled synthetic test cases | 14.8 | 21.1 |

Table 8: The pass rate evaluated by executing the annotated ground-truth solution program on our synthetic test cases. The *Model* column denotes the one using the corresponding test cases for preference optimization.

| Model | Pass Rate |
| --- | --- |
| PFPO-LLM Iter. 0 (APPs) | **82.41** |
| PFPO-Self Iter. 0 (APPs) | 62.68 |
| PFPO-Self Iter. 1 (APPs & M.C.) | 66.80 |
| PFPO-Self Iter. 2 (APPs & M.C. & xCode.) | 66.75 |

Table 9: Overall results on mathematical reasoning benchmarks. The indentation across different levels represents the corresponding model is initialized from the parent-level. *M.S.* refers to the short of *MathScale*, and *S.C.* means *Self-Consistency*.

| | MATH | GSM8K | College Math |
|---|---|---|---|
| Llama-3.1-8B-base | | | |
| w/ SFT (M.S.-500k) | 53.8 | 85.1 | 34.6 |
| w/ DPO (M.S.-500k) | 53.8 | 86.0 | 35.1 |
| w/ pDPO (M.S.-500k) | 55.0 | 86.6 | 35.8 |
| w/ pDPO (Numina-S.C. 160k) | 55.9 | 87.6 | 36.6 |
| w/ pDPO (Numina-S.C. 320k) | 56.6 | 88.9 | 37.0 |
| w/ pDPO (Numina-S.C. 480k) | 57.0 | 88.8 | 36.7 |
| w/ pDPO (Numina-S.C. 640k) | 57.4 | 89.1 | 37.6 |
| w/ pDPO (Numina-S.C. 790K) | **57.8** | **89.6** | **38.0** |
| Mathstral-7B-v0.1 | 58.3 | 85.6 | 34.3 |
| w/ SFT (M.S.-500k) | 61.4 | 87.3 | 38.4 |
| w/ DPO (M.S.-500k) | 63.0 | 88.6 | 39.1 |
| w/ pDPO (M.S.-500k) | 66.7 | 90.0 | 41.3 |
| w/ pDPO (M.S.-300k-S.C., Iter. 0) | 67.8 | **90.8** | 42.0 |
| w/ pDPO (M.S.-300k-S.C., Iter. 1) | **68.6** | 90.3 | 42.2 |
| w/ pDPO (M.S.-300k-S.C., Iter. 2) | 68.2 | 90.4 | **42.3** |

Table 10: The performance on the more challenging sub-test set of MATH, which is constructed from the failed questions Mathstral-7B w/ SFT via Maj@16.

| Mathstral-7B | Greedy Decoding |
|---|---|
| w/ SFT | 19.6 |
| PFPO-LLM Iter. 0 | 23.6 |
| PFPO-Self Iter. 1 | 24.6 |
| PFPO-Self Iter. 2 | 26.7 |

Table 11: Overall results (Pass@1) on program generation benchmarks. *M.C.* refers to the prompt set of Magicoder (Wei et al., 2024), and *xCode.* is the short for xCodeEval (Khan et al., 2024b). *Introductory*, *Interview*, and *Competition* indicate the three difficulty levels of APPs. *S.C.* indicates that the test cases used for constructing preference dataset are synthesized through self-consistency. On the contrary, the row without *S.C.* refers that the test cases the golden ones from the original dataset.

| | APPs | | | | HumanEval | MBPP |
|---|---|---|---|---|---|---|
| | Overall | Introductory | Interview | Competition | | |
| Deepseek-coder-v1.5-chat | 14.3 | 35.7 | 10.8 | 3.2 | 75.6 | 73.9 |
| w/ SFT (APPs - GPT-4o) | 15.4 | 37.8 | 11.6 | 4.1 | 72.0 | 72.8 |
| w/ DPO (APPs) | 16.3 | 36.2 | 13.3 | 5.3 | 74.4 | 74.3 |
| w/ pDPO (APPs) | 16.9 | 37.3 | 13.8 | 6.1 | 73.8 | 73.2 |
| w/ DPO (APPs - S.C.) | 16.8 | 39.2 | 13.2 | 5.3 | 76.2 | 75.9 |
| w/ pDPO (APPs - S.C.) | 17.4 | 37.5 | 14.8 | 5.4 | 73.2 | 75.1 |
| w/ DPO (APPs & M.C. - S.C.) | 18.0 | 39.2 | 14.9 | 6.2 | **79.3** | 75.5 |
| w/ DPO (APPs & M.C. & xCode. - S.C.) | **19.2** | **42.2** | 15.8 | 6.5 | 75.0 | 74.2 |
| w/ pDPO (APPs & M.C. & xCode. - S.C.) | 19.1 | 40.9 | **15.9** | **6.9** | 73.8 | 75.1 |
| w/ pDPO (APPs & M.C. - S.C.) | 18.5 | 40.0 | 15.3 | 6.5 | 75.6 | **76.3** |

Table 12: Overall results on LiveCodeBench. We follow the recommended setting by sampling 10 solutions for each problem with temperature as 0.2, and estimating the Pass@1 results. The cutoff date of the test questions is from **2023-09-01** to **2024-09-01**. All results except those of our models are referenced from the official leaderboard.

|  | Overall | Easy | Medium | Hard |
|---|---|---|---|---|
| Claude-3.5-Sonnet | 51.3 | 87.2 | 45.3 | 11.0 |
| DeepSeekCoder-V2 | 41.9 | 79.9 | 32.0 | 4.9 |
| Claude-3-Sonnet | 26.9 | 67.2 | 7.3 | 1.4 |
| Claude-3-Haiku | 24.0 | 61.3 | 5.5 | 0.9 |
| GPT-3.5-Turbo-0125 | 24.0 | 55.0 | 11.6 | 0.3 |
| LLama-3-70b-Instruct | 27.4 | 59.4 | 15.6 | 1.3 |
| Deepseek-Coder-33b-Chat | 23.4 | 56.1 | 8.6 | 0.9 |
| Deepseek-coder-v1.5-chat | 21.1 | 51.3 | 7.4 | 0.2 |
| w/ SFT (APPs - GPT-4o) | 22.9 | 53.0 | 10.6 | 0.2 |
| w/ DPO (APPs) | 22.9 | 53.7 | 9.4 | 1.0 |
| w/ pDPO (APPs) | 22.9 | 55.0 | 8.1 | 1.3 |
| w/ DPO (APPs - S.C.) | 24.2 | 55.7 | 11.0 | 0.8 |
| w/ pDPO (APPs - S.C.) | 23.4 | 54.2 | 10.3 | 0.7 |
| w/ DPO (APPs & M.C. - S.C.) | 23.7 | 55.8 | 9.5 | 1.1 |
| w/ DPO (APPs & M.C. & xCode. - S.C.) | 23.7 | 55.8 | 9.4 | 1.3 |
| w/ pDPO (APPs & M.C. & xCode. - S.C.) | 24.3 | 56.8 | 9.8 | **1.6** |
| w/ pDPO (APPs & M.C. - S.C.) | **24.6** | **56.9** | **11.4** | 0.2 |

You are an expert programmer. Your task is to write some test cases to the programming problems to help verify the expected program solutions. You only need to give me the inputs in the required format. Now, let me introduce the details to you:

## Program Format

There will be two kinds of programming problems. One type of problem accepts standard input-output stream. As a result, the test case inputs should contain only the inputs text stream.

Another kind of problem is based on function calling, which shows a segment of starter code to illustrate the function head, defining the name of the arguments to be accepted. In this case, you should return me the inputs in the format of function callinåg, like `function_name(*arguments)`.

## Response Format

You should return me the test case inputs in `json_object` format. You need to generate **10** groups of test case inputs, and each key field is named as `test_case_i`, where `i` is the index of the test case. The value of each key is the test case inputs in the required format, which should be a string.

## Examples for Standard Input-Output and Function Calling.

### Standard Input-Output Stream

#### Programming Problem

Polycarp has $n$ different binary words. A word called binary if it contains only characters '0' and '1'. For example, these words are binary: "0001", "11", "0" and "0011100".

Polycarp wants to offer his set of $n$ binary words to play a game "words". In this game, players name words and each next word (starting from the second) must start with the last character of the previous word. The first word can be any. For example, these sequence of words can be named during the game: "0101", "1", "10", "00", "00001".

Word reversal is the operation of reversing the order of the characters. For example, the word "0111" after the reversal becomes "1110", the word "11010" after the reversal becomes "01011".

Probably, Polycarp has such a set of words that there is no way to put them in the order correspondent to the game rules. In this situation, he wants to reverse some words from his set so that: the final set of $n$ words still contains different words (i.e. all words are unique); there is a way to put all words of the final set of words in the order so that the final sequence of $n$ words is consistent with the game rules.

Polycarp wants to reverse minimal number of words. Please, help him.

-----Input-----

The first line of the input contains one integer $t$ ($1 \le t \le 10^4$) — the number of test cases in the input. Then $t$ test cases follow.

The first line of a test case contains one integer $n$ ($1 \le n \le 2\cdot10^5$) — the number of words in the Polycarp's set. Next $n$ lines contain these words. All of $n$ words aren't empty and contains only characters '0' and '1'. The sum of word lengths doesn't exceed $4\cdot10^6$. All words are different.

Guaranteed, that the sum of $n$ for all test cases in the input doesn't exceed $2\cdot10^5$. Also, guaranteed that the sum of word lengths for all test cases in the input doesn't exceed $4\cdot10^6$.

-----Output-----

Print answer for all of $t$ test cases in the order they appear.

Figure 6: 2-shot prompt for test case inputs generation. Page 1.

If there is no answer for the test case, print -1. Otherwise, the first line of the output should contain $k$ ($0 \le k \le n$) — the minimal number of words in the set which should be reversed. The second line of the output should contain $k$ distinct integers — the indexes of the words in the set which should be reversed. Words are numerated from $1$ to $n$ in the order they appear. If $k=0$ you can skip this line (or you can print an empty line). If there are many answers you can print any of them.

-----Example-----
Input
4
4
0001
1000
0011
0111
3
010
101
0
2
00000
00001
4
01
001
0001
00001

Output
1
3
-1
0

2
1 2

#### Response

```
{
    "test_case_0": "3\n3\n101\n110\n011\n2\n01\n10\n4\n0001\n1000\n0011\n0111",
    "test_case_1": "2\n2\n01\n10\n3\n000\n111\n110",
    ...
}
```

### Function Calling

#### Programming Problem

Given a single positive integer x, we will write an expression of the form x (op1) x (op2) x (op3) x ... where each operator op1, op2, etc. is either addition, subtraction, multiplication, or division (+, -, *, or /). For example, with x = 3, we might write 3 * 3 / 3 + 3 - 3 which is a value of 3.
When writing such an expression, we adhere to the following conventions:

The division operator (/) returns rational numbers.
There are no parentheses placed anywhere.
We use the usual order of operations: multiplication and division happens before addition and subtraction.
It's not allowed to use the unary negation operator (-). For example, "x - x" is a valid expression as it only uses subtraction, but "-x + x" is not because it uses negation.

Figure 7: 2-shot prompt for test case inputs generation. Page 2.

We would like to write an expression with the least number of operators such that the expression equals the given target. Return the least number of operators used.

Example 1:
Input: x = 3, target = 19
Output: 5
Explanation: 3 * 3 + 3 * 3 + 3 / 3. The expression contains 5 operations.

Example 2:

Input: x = 5, target = 501
Output: 8
Explanation: 5 * 5 * 5 * 5 - 5 * 5 * 5 + 5 / 5. The expression contains 8 operations.

Example 3:
Input: x = 100, target = 100000000
Output: 3
Explanation: 100 * 100 * 100 * 100. The expression contains 3 operations.

Note:

$2 <= x <= 100$
$1 <= target <= 2 * 10^8$

```
class Solution:
    def leastOpsExpressTarget(self, x: int, target: int) -> int:
```

#### Response

```
{
    "test_case_0": "leastOpsExpressTarget(3, 19)",
    "test_case_1": "leastOpsExpressTarget(3, 32)",
    "test_case_2": "leastOpsExpressTarget(6, 100)",
    ...
}
```

## Get Started

Note that in the above examples, I omit some test case inputs. You should return **10** groups of inputs to me in `json_object` format.

#### Programming Problem

[[Question]]

#### Response

Figure 8: 2-shot prompt for test case inputs generation. Page 3.

You are an expert programmer. I will show you a programming problem. Please carefully comprehend the requirements in the problem, and write down the solution program to pass it under the given time and memory constraints.

**REMEMBER** to strictly follow the steps below to help reduce the potential flaws:
(1) According to the input scale and the time/memory constraints, think about the time complexity and space complexity of your solution.
(2) Think **step-by-step** to design the algorithm.
(3) Translate your thoughts into Python program to solve it.

Besides, your Python solution program should be located between <BEGIN> and <END> tags:
<BEGIN>
t = int(input())
...
print(ans)
<END>

Here is an example:

## Problem

You are given an array $a$ of length $n$ consisting of zeros. You perform $n$ actions with this array: during the $i$-th action, the following sequence of operations appears: Choose the maximum by length subarray (continuous subsegment) consisting only of zeros, among all such segments choose the leftmost one; Let this segment be $[l; r]$. If $r-l+1$ is odd (not divisible by $2$) then assign (set) $a[\frac{l+r}{2}] := i$ (where $i$ is the number of the current action), otherwise (if $r-l+1$ is even) assign (set) $a[\frac{l+r-1}{2}] := i$.

Consider the array $a$ of length $5$ (initially $a=[0, 0, 0, 0, 0]$). Then it changes as follows: Firstly, we choose the segment $[1; 5]$ and assign $a[3] := 1$, so $a$ becomes $[0, 0, 1, 0, 0]$; then we choose the segment $[1; 2]$ and assign $a[1] := 2$, so $a$ becomes $[2, 0, 1, 0, 0]$; then we choose the segment $[4; 5]$ and assign $a[4] := 3$, so $a$ becomes $[2, 0, 1, 3, 0]$; then we choose the segment $[2; 2]$ and assign $a[2] := 4$, so $a$ becomes $[2, 4, 1, 3, 0]$; and at last we choose the segment $[5; 5]$ and assign $a[5] := 5$, so $a$ becomes $[2, 4, 1, 3, 5]$.

Your task is to find the array $a$ of length $n$ after performing all $n$ actions. Note that the answer exists and unique.

You have to answer $t$ independent test cases.

-----Input-----

The first line of the input contains one integer $t$ ($1 \le t \le 10^4$) — the number of test cases. Then $t$ test cases follow.

The only line of the test case contains one integer $n$ ($1 \le n \le 2 \cdot 10^5$) — the length of $a$.

It is guaranteed that the sum of $n$ over all test cases does not exceed $2 \cdot 10^5$ ($\sum n \le 2 \cdot 10^5$).

-----Output-----

For each test case, print the answer — the array $a$ of length $n$ after performing $n$ actions described in the problem statement. Note that the answer exists and unique.

Figure 9: 1-shot competition-level code generation prompt. When being applied to SFT, the 1-shot example is removed. Page 1.

If there is no answer for the test case, print -1. Otherwise, the first line of the output should contain $k$ ($0 \le k \le n$) — the minimal number of words in the set which should be reversed. The second line of the output should contain $k$ distinct integers — the indexes of the words in the set which should be reversed. Words are numerated from $1$ to $n$ in the order they appear. If $k=0$ you can skip this line (or you can print an empty line). If there are many answers you can print any of them.

-----Example-----
Input
4
4
0001
1000
0011
0111
3
010
101
0
2
00000
00001
4
01
001
0001
00001

Output
1
3
-1
0

2
1 2

#### Response

```
{
    "test_case_0": "3\n3\n101\n110\n011\n2\n01\n10\n4\n0001\n1000\n0011\n0111",
    "test_case_1": "2\n2\n01\n10\n3\n000\n111\n110",
    ...
}
```

### Function Calling

#### Programming Problem

Given a single positive integer x, we will write an expression of the form x (op1) x (op2) x (op3) x ... where each operator op1, op2, etc. is either addition, subtraction, multiplication, or division (+, -, *, or /). For example, with x = 3, we might write 3 * 3 / 3 + 3 - 3 which is a value of 3.
When writing such an expression, we adhere to the following conventions:

The division operator (/) returns rational numbers.
There are no parentheses placed anywhere.
We use the usual order of operations: multiplication and division happens before addition and subtraction.
It's not allowed to use the unary negation operator (-). For example, "x - x" is a valid expression as it only uses subtraction, but "-x + x" is not because it uses negation.

Figure 10: 1-shot competition-level code generation prompt. When being applied to SFT, the 1-shot example is removed. Page 2.

We would like to write an expression with the least number of operators such that the expression equals the given target. Return the least number of operators used.

Example 1:
Input: x = 3, target = 19
Output: 5
Explanation: 3 * 3 + 3 * 3 + 3 / 3. The expression contains 5 operations.

Example 2:

Input: x = 5, target = 501
Output: 8
Explanation: 5 * 5 * 5 * 5 - 5 * 5 * 5 + 5 / 5. The expression contains 8 operations.

Example 3:
Input: x = 100, target = 100000000
Output: 3
Explanation: 100 * 100 * 100 * 100. The expression contains 3 operations.

Note:

2 <= x <= 100
1 <= target <= 2 * 10^8

class Solution:
    def leastOpsExpressTarget(self, x: int, target: int) -> int:

#### Response

{
    "test_case_0": "leastOpsExpressTarget(3, 19)",
    "test_case_1": "leastOpsExpressTarget(3, 32)",
    "test_case_2": "leastOpsExpressTarget(6, 100)",
    ...
}

## Get Started

Note that in the above examples, I omit some test case inputs. You should return **10** groups of inputs to me in `json_object` format.

#### Programming Problem

[[Question]]

#### Response

Figure 11: 1-shot competition-level code generation prompt. When being applied to SFT, the 1-shot example is removed. Page 3.

