# OpenReview forum: "Preference Optimization for Reasoning with Pseudo Feedback"
_ICLR.cc/2025/Conference — ICLR 2025 Spotlight_

### Official Review · Reviewer_dJUL · 2024-11-01

**Soundness:** 3
**Presentation:** 3
**Contribution:** 3
**Rating:** 6
**Confidence:** 4

**Summary:**

This paper proposes Pseudo-Feedback Preference Optimization (PFPO), a novel approach for preference optimization in large language models (LLMs) tailored to reasoning tasks in mathematical and coding domains. PFPO sidesteps the dependency on human-verified labels by generating pseudo feedback through test cases, either created by frontier LLMs or using self-consistency from the policy LLM itself. Using Llama-3.1-8B/Mathstral-7B for mathematical reasoning and Deepseek-coder-7B-v1.5 for code generation, PFPO achieved notable improvements on the MATH, GSM8K, College Math, and LiveCodeBench benchmarks.

**Strengths:**

1. Comprehensive experiments across a range of reasoning tasks and LLM models.
2. Clear presentation and good formalization.
3. Empirical results show the method’s effectiveness.

**Weaknesses:**

1. Limited Novelty at ICLR Level:
   The technique of leveraging unit tests as feedback for direct preference optimization (DPO/PPO) is already established in existing research. Previous works, such as CodeRL [1], and more recent studies [2][3], have incorporated unit tests and compiler feedback within reinforcement learning (RL)/PPO frameworks for code generation. Similarly, fine-grained feedback with DPO has been applied in mathematical reasoning [4], and LLM-based feedback with DPO has been explored for code generation [5]. Although PFPO Self-iter demonstrates strong performance, its approach of utilizing LLM-generated feedback signals for DPO closely aligns with these prior works, which limits its methodological originality.

   - [1] Le, H., Wang, Y., Gotmare, A. D., Savarese, S., & Hoi, S. C. H. (2022). CodeRL: Mastering code generation through pretrained models and deep reinforcement learning. *Advances in Neural Information Processing Systems*, 35, 21314-21328.
   - [2] Liu, J., Zhu, Y., Xiao, K., Fu, Q., Han, X., Yang, W., & Ye, D. (2023). Rltf: Reinforcement learning from unit test feedback. arXiv preprint arXiv:2307.04349.
   - [3] Dou, S., et al. (2024). Stepcoder: Improve code generation with reinforcement learning from compiler feedback. arXiv preprint arXiv:2402.01391.
   - [4] Lai, X., et al. (2024). Step-dpo: Step-wise preference optimization for long-chain reasoning of LLMs. arXiv preprint arXiv:2406.18629.
   - [5] Weyssow, M., Kamanda, A., & Sahraoui, H. (2024). CodeUltraFeedback: An LLM-as-a-Judge Dataset for Aligning Large Language Models to Coding Preferences. arXiv preprint arXiv:2403.09032.

2. Insufficient Discussion of Limitations in Self-Generated Pseudo Labels:
   The paper could benefit from a more thorough examination of the potential biases and risks associated with self-generated pseudo labels, such as model bias and the tendency to overfit synthetic feedback. A clearer analysis would provide a more robust perspective on PFPO's limitations. Additionally, connecting this limitation to the plateau effect noted in Question 1 would be insightful, as similar issues with DPO setups have been documented, as seen in [4]. Empirical evaluation or discussion on the effects of varying types and quantities (quesiton 3) of synthetic feedback could strengthen this analysis further.

**Questions:**

1. PFPO with self-iteration clearly plateaus after several iterations. Can the authors provide more insights on what stage causes this, such as whether the quality of generated test cases plateaus or the policy model stops learning?
2. In Table 1, the baseline score for Llama-3.1-8B-base seems missing. Could this be clarified?
3. In section 4.2.2, synthetic test cases outperform ground truth, likely due to greater quantity. Since generated inputs and ground truth programs are available, why not run the programs on generated inputs to match the number of ground truth test cases, enabling a fairer comparison?

---

> ### Author Response · Authors · 2024-11-22
> **Response - 1 / 4**
>
> > Limited Novelty at ICLR Level:
> >
> >The technique of leveraging unit tests as feedback for direct preference optimization (DPO/PPO) is already established in existing research. Previous works, such as CodeRL [1], and more recent studies [2][3], have incorporated unit tests and compiler feedback within reinforcement learning (RL)/PPO frameworks for code generation. Similarly, fine-grained feedback with DPO has been applied in mathematical reasoning [4], and LLM-based feedback with DPO has been explored for code generation [5]. Although PFPO Self-iter demonstrates strong performance, its approach of utilizing LLM-generated feedback signals for DPO closely aligns with these prior works, which limits its methodological originality.
> >
> > [1] Le, H., Wang, Y., Gotmare, A. D., Savarese, S., & Hoi, S. C. H. (2022). CodeRL: Mastering code generation through pretrained models and deep reinforcement learning. Advances in Neural Information Processing Systems, 35, 21314-21328.
> >
> > [2] Liu, J., Zhu, Y., Xiao, K., Fu, Q., Han, X., Yang, W., & Ye, D. (2023). Rltf: Reinforcement learning from unit test feedback. arXiv preprint arXiv:2307.04349.
> >
> > [3] Dou, S., et al. (2024). Stepcoder: Improve code generation with reinforcement learning from compiler feedback. arXiv preprint arXiv:2402.01391.
> >
> > [4] Lai, X., et al. (2024). Step-dpo: Step-wise preference optimization for long-chain reasoning of LLMs. arXiv preprint arXiv:2406.18629.
> >
> > [5] Weyssow, M., Kamanda, A., & Sahraoui, H. (2024). CodeUltraFeedback: An LLM-as-a-Judge Dataset for Aligning Large Language Models to Coding Preferences. arXiv preprint arXiv:2403.09032.
>
>
> ### **On Novelty**
> Thank you for your thoughtful comments and for highlighting relevant prior works[1][2][3][4][5]. We would like to take this opportunity to clarify our contributions and differentiate them from the studies you mentioned.
>
> The methods designed in [1][2][3] are all for the *coding* task leveraging unit tests and compiler feedback to guide RL training. Specifically, Le et al. [1] employed example unit tests (input-output pairs provided in problem descriptions) to produce feedback for the actor-critic RL approach, while Liu et al. [2] designed more fine-grained feedback and employed online RL method. Dou et al., [3] mainly focused on employing test cases to create step-level feedback.
>
> However, all studies relied on **human annotated test cases**, which are challenging to obtain for large-scale training datasets. For example, the Magicoder dataset lacks test case annotations. In contrast, our work leverages *synthetic test cases* without expensive human annotations, which can be easily created on a larger scale. Specifically, we differ from [1][2][3] in the following aspects:
> - We propose a method for synthesizing both test case inputs and outputs.
> - We verify the feasibility of using self-constructed test cases for iterative improvement in code generation.
> - To the best of our knowledge, we are the first to explore synthetic pseudo-feedback via test cases beyond solving coding problems. We extend our method to other reasoning tasks such as mathematical reasoning.
>
> Lai et al., [4] propose to use GPT-4 for error location to provide step-level supervision. In the error location step, **gold answer labels** are employed to collect solutions with errors. On the other hand, we create pseudo answer labels and our method can be applied to coding in addition to math reasoning.
>
> The work [4] is more directly related to the pDPO [6] method we employed for preference optimization. Also note with the pseudo feedback we created, we can incorporate different preference learning and RL methods such as DPO, pDPO, stepDPO and PPO. And employing DPO or pDPO in our method is not our contribution.

---

> ### Author Response · Authors · 2024-11-22
> **Response - 2 / 4**
>
> Finally, Weyssow et al. [5] relied on frontier LLMs to annotate preference pairs of code solutions using an LLM-as-a-judge approach. Our approach is different. We employ synthetic test case-based program verification to create preference pairs, which is more sample efficient. Once the test cases of a problem are created, we can use them to construct multiple preference pairs, while [5] requires frontier LLM to annotate each pair of sampled solutions from the policy model. Besides, the test case based verification enables on-policy learning, e.g., iterative DPO or PPO. Using frontier LLMs to annotate solutions can only be used to train a reward model or DPO model in single iteration.
>
> In summary, while previous works focused on leveraging feedback (either from manually annotated unit tests or LLM-as-a-judge), our contributions lie in:
> - Learning from pseudo-feedback generated both by frontier LLMs and self-consistency.
> - Developing a scalable framework for pseudo-feedback construction, enabling large-scale DPO training for both math and coding domains.
>
>
> We appreciate your feedback and will incorporate a more thorough comparison with related works in the revised manuscript.
>
> [1] Le, H., Wang, Y., Gotmare, A. D., Savarese, S., & Hoi, S. C. H. (2022). CodeRL: Mastering code generation through pretrained models and deep reinforcement learning. Advances in Neural Information Processing Systems, 35, 21314-21328.
> [2] Liu, J., Zhu, Y., Xiao, K., Fu, Q., Han, X., Yang, W., & Ye, D. (2023). Rltf: Reinforcement learning from unit test feedback. arXiv preprint arXiv:2307.04349.
> [3] Dou, S., et al. (2024). Stepcoder: Improve code generation with reinforcement learning from compiler feedback. arXiv preprint arXiv:2402.01391.
> [4] Lai, X., et al. (2024). Step-dpo: Step-wise preference optimization for long-chain reasoning of LLMs. arXiv preprint arXiv:2406.18629.
> [5] Weyssow, M., Kamanda, A., & Sahraoui, H. (2024). CodeUltraFeedback: An LLM-as-a-Judge Dataset for Aligning Large Language Models to Coding Preferences. arXiv preprint arXiv:2403.09032.
> [6] Fangkai Jiao, Chengwei Qin, Zhengyuan Liu, Nancy F. Chen, Shafiq Joty (2024). Learning Planning-based Reasoning by Trajectories Collection and Process Reward Synthesizing

---

> ### Author Response · Authors · 2024-11-22
> **Response - 3 / 4**
>
> > PFPO with self-iteration clearly plateaus after several iterations. Can the authors provide more insights on what stage causes this, such as whether the quality of generated test cases plateaus or the policy model stops learning?
>
> Thanks for your insightful comments! To investigate why “PFPO self” plateaus after several iterations, we first assess the quality of generated test cases as you recommended.
>
> ### **Quality of Generated Test-cases**
> We choose to conduct our experiments on coding, since APPs is part of our training dataset with gold program solution annotation (see line 300), which can be used to verify our synthetic test cases for each iteration.
>
> Specifically, for each question in the APPs training set, we selected a ground-truth program to evaluate the synthetic test cases. A synthetic test case is valid, if and only if given the test case input and its output matches the output of the ground-truth program.
> We calculate the "Test Case Pass Rate," which represents the percentage of valid test cases among the total generated.
> The results are as follows, where the `Model` column denotes the model optimized by the pseudo feedback constructed from the corresponding test cases.
>
> | Model                                 | Test Case Pass Rate |
> |--------------------------------------|:---------------------:|
> | PFPO-Self Iter. 0 (APPs)             | 62.68              |
> | PFPO-Self Iter. 1 (APPs & M.C.)      | 66.80              |
> | PFPO-Self Iter. 2 (APPs & M.C. & xCode.) | 66.75          |
>
> Indeed, the quality of synthetic test cases of Iter. 1 is the best and Iter. 2 is slightly worse and the final coding performance follows a similar trend (see Table 3). We assume that the quality of synthetic test cases can be one of the contributing factors to the marginal improvement over iterations. We will add these analyses to our updated manuscript.
>
> ### **Why Plateaus Occur**
>
> As also observed in [1][2], iterative style DPO tends to saturate after several iterations, regardless of whether the preference pairs are human labeled or synthetic. We believe there are at least two reasons for the saturation. The first one (as mentioned earlier) is the quality of synthetic test cases may plateau at some point.
>
> The second reason could be that as the model improves with each iteration, more problems in the training set become "too easy" for the model at its current stage. For extremely challenging problems where no correct solution can be generated despite many attempts (sampling), we are unable to create valid preference pairs and are thus unlikely to solve them in the next iteration. The number of moderately challenging problems, which are most effective for improving the model, gradually decreases after each iteration.
>
> **Potential Solutions:** One possible solution to this is introducing unseen problems in each iteration to increase the likelihood of hitting new moderately challenging problems.
>
> In the third block of Table 1 (line 336 to 342), we simulated the above setting by introducing new problems across iterations. As detailed in Appendix A, we split the NuminaMath dataset into five subsets of equal size and use a different subset for each iteration. We observed consistent improvements across iterations (see w/ PFPO-Self Iter. 1 to 5 rows in Table 1).
>
> However, please note that applying the above method requires a significant amount of unseen high-quality questions in that domain. Therefore, we cannot apply this strategy with “Mathstral-7B-v0.1” based models (line 343 to 347; Mathstral-7B-v0.1 has been trained on Numina already) or in coding domain (high quality prompts in coding domain is very limited).
>
> Finally, it is worth noting that our method’s advantage lies in its scalability, as no additional human annotation for answers is required when incorporating new problems.
>
> [1] Iterative Preference Learning from Human Feedback: Bridging Theory and Practice for RLHF under KL-constraint
> [2] Bootstrapping Language Models with DPO Implicit Rewards
>
> Thanks again for your insightful question. All results and discussions above will be added to our updated manuscript.

---

> ### Author Response · Authors · 2024-11-22
> **Response - 4 / 4**
>
> > In Table 1, the baseline score for Llama-3.1-8B-base seems missing. Could this be clarified?
>
> Thank you for pointing this out. We will update Table 1 to include the baseline scores for Llama-3.1-8B-base:
>
> |                       | MATH (4-shot) | GSM8K (8-shot) | College Math (4-shot) |
> |-----------------------|:---------------:|:----------------:|:-----------------------:|
> | Llama-3.1-8B-base     | 20.3          | 56.7           | 20.12                |
>
> > In section 4.2.2, synthetic test cases outperform ground truth, likely due to greater quantity. Since generated inputs and ground truth programs are available, why not run the programs on generated inputs to match the number of ground truth test cases, enabling a fairer comparison?
>
> Thank you for highlighting this important point!
>
> We downsampled the synthetic test cases to align with the number of ground truth test cases. Specifically, for each coding problem, if the synthesized test cases exceeded the number of ground truth test cases, we randomly selected a subset of synthesized cases to match the number of ground truth test cases. Otherwise, all synthetic test cases were retained.
>
> The results are as follows:
>
> | DS-Coder-v1.5-chat                        | APPs  | LiveCodeBench |
> |-------------------------------------------|:-------:|:---------------:|
> | w/ SFT                                    | 15.4  | 22.9          |
> | w/ golden test cases                      | 16.9  | 22.9          |
> | w/ synthetic test cases                   | **17.4**  | **23.4**          |
> | w/ down-sampled synthetic test cases      | 14.8  | 21.1          |
>
>
> These results demonstrate that reducing the number of synthetic test cases significantly decreases performance. This highlights the importance of scaling test cases for verifying program correctness effectively.
>
> Finally, we sincerely appreciate your valueable comments. We will update all the experimental results and analysis to our revised manuscript.

---

> > ### Comment · Reviewer_dJUL · 2024-11-22
> >
> > Thank you for your detailed and thoughtful responses. I appreciate the effort in addressing my questions and providing additional experimental results.
> >
> > The responses sufficiently addressed my concerns, and the new results further substantiated the claims made in the paper.
> > Based on the clarifications and supporting evidence, I have revised my rating from 5 to 6, contribution 2 to 3.

---

> > > ### Author Response · Authors · 2024-11-23
> > >
> > > Thank you for taking the time to review our responses! We truly appreciate your positive reassessment and for raising the score!

---

### Official Review · Reviewer_NHWB · 2024-11-04

**Soundness:** 4
**Presentation:** 4
**Contribution:** 3
**Rating:** 8
**Confidence:** 3

**Summary:**

This paper introduces PFPO (Pseudo-Feedback Preference Optimization) that creates pseudo feedback for reasoning tasks by framing the labeling of solutions to reason problems as an evaluation against associated test cases. They explore two forms of pseudo feedback based on test cases: one generated by frontier LLMs and the other by extending self-consistency to multi-test-case. Experiments are conducted on math reasoning and code generation and observe performance improvements.

**Strengths:**

* The paper presents a novel approach to preference optimization by leveraging pseudo feedback.
* The experiments are well-designed and comprehensive, covering both mathematical reasoning and coding tasks. The results show substantial improvements over baseline models and even surpass some state-of-the-art models, indicating the high quality of the proposed methods.
* The paper is well-structured and clearly written. The methodology is explained in detail, and the experimental setup and results are presented in a way that is easy to follow.

**Weaknesses:**

Please see the questions below.

**Questions:**

* The method relies on frontier LLMs to generate pseudo feedback. How does the performance of the proposed method degrade if the frontier LLMs are not available or their performance is suboptimal?
* How does the quality of pseudo feedback (from both frontier LLMs and self-consistency) impact the final performance of the LLM? Is there a way to measure or improve this quality?
* Could the authors provide more details on the computational requirements and potential optimizations for the iterative training process?

---

> ### Author Response · Authors · 2024-11-22
> **Response - 1 / 2**
>
> > The method relies on frontier LLMs to generate pseudo feedback. How does the performance of the proposed method degrade if the frontier LLMs are not available or their performance is suboptimal?
>
> Thank you for raising this important question.
> To investigate this, we compared the results of PFPO at Iter.0 with LLM feedback (gpt-4o) and with self-consistency feedback (see below) on the APPs training dataset:
>
> | DS-Coder-v1.5-chat            | APPs  | LiveCodeBench |
> |-------------------------------|:-------:|:---------------:|
> | w/ PFPO-LLM Iter. 0 (APPs)    | **17.9**  | **24.0**          |
> | w/ PFPO-Self Iter. 0 (APPs)   | 17.4  | 23.4          |
>
>
>
> From the table, we observe that feedback generated by the frontier LLM leads to performance improvements on APPs compared to self-generated pseudo feedback. This highlights the value of leveraging frontier LLMs for higher-quality feedback.
>
> > How does the quality of pseudo feedback (from both frontier LLMs and self-consistency) impact the final performance of the LLM? Is there a way to measure or improve this quality?
>
>
> In general, higher-quality feedback produces better results.
>
> ### **On Measuring Quality of Pseudo Feedback**
>
> We choose to conduct our experiments on coding, since APPs is part of our training dataset with gold program solution annotation (see line 300), which can be used to verify our synthetic test cases for each iteration.
>
> Specifically, for each question in the APPs training set, we selected a ground-truth program to evaluate the synthetic test cases. A synthetic test case is valid, if and only if given the test case input and its output matches the output of the ground-truth program.
>
> We calculate the "Test Case Pass Rate," which represents the percentage of valid test cases among the total generated. Higher Test Case Pass Rate means better pseudo feedback quality.
>
> The results are as follows, where the `Model` column denotes the model optimized by the pseudo feedback constructed from the corresponding test cases.
>
> | Model                                 | Test Case Pass Rate |
> |--------------------------------------|:---------------------:|
> | PFPO-LLM Iter. 0 (APPs)              | **82.41**              |
> | PFPO-Self Iter. 0 (APPs)             | 62.68              |
> | PFPO-Self Iter. 1 (APPs & M.C.)      | **66.80**              |
> | PFPO-Self Iter. 2 (APPs & M.C. & xCode.) | 66.75          |
>
> However, we found it challenging to directly measure the pseudo feedback quality on the math domain, since we use synthetic math datasets in experiments and gold answer labels are missing in our datasets.
>
> ### **Impact of Pseudo Feedback Quality on Final Performance**
>
> By comparing the "Test Case Pass Rate" from the previous table with PFPO's final performance on APPs and LiveCodeBench in the table below, we observe that higher-quality pseudo feedback generally results in better downstream task results. This trend is evident when comparing PFPO-LLM Iter. 0 with PFPO-Self Iter. 0, as well as comparing PFPO-Self Iter. 0 with Iter. 1, and Iter. 2.
>
> | DS-Coder-v1.5-chat                 | APPs  | LiveCodeBench |
> |------------------------------------|-------|---------------|
> | w/ PFPO-LLM Iter. 0 (APPs)         | 17.9  | 24.0          |
> | w/ PFPO-Self Iter. 0 (APPs)        | 17.4  | 23.4          |
> | w/ PFPO-Self Iter. 1 (APPs & M.C.) | 18.0  | 23.7          |
> | w/ PFPO-Self Iter. 2 (APPs & M.C. & xCode.) | **19.1** | **24.3**   |
>
> However, an exception exists: although the pseudo feedback quality of PFPO-LLM Iter. 0 is significantly higher than that of PFPO-Self Iter. 2, its final performance is worse. This is likely attributed to the off-policy learning. Using Iterative DPO or online RL algorithms may help this.
>
> Based on the results above, therefore, we recommend using the quality of pseudo feedback as an indicator of overfitting in your training set (or a small development set). If the pseudo feedback quality no longer improves, it may serve as a strong signal to halt further training of your model with pseudo feedback.
>
> ### **On Improving Pseudo Feedback Quality**
>
> A simple approach to enhance pseudo feedback quality is to generate more samples from the LLM (or self-consistency), or combine pseudo feedback from different checkpoints (ensemble). However, this comes with the trade-off of increased computational costs.

---

> ### Author Response · Authors · 2024-11-22
> **Response - 2 / 2**
>
> > Could the authors provide more details on the computational requirements and potential optimizations for the iterative training process?
>
> In the math reasoning experiment using Mathstral-7B-v0.1, we conducted PFPO on the MathScale-300K dataset. Using 8×A100-80G GPUs with NVLink, each iteration of PFPO training takes approximately 6 hours. For creating the self-consistency based pseudo feedback and the DPO pairs, we need to do inference on our policy models a lot. It requires around 20 hours to finish the solution sampling for MathScale-300K using vLLM with 8xA100-40G GPUs.
>
> The coding experiments are conducted on a much smaller scale, resulting in significantly lower computational costs.
>
> Creating DPO pairs (or SC based pseudo feedback) is more computationally intensive than the DPO training itself. However, the process is highly parallelizable. For inference with a 7B/8B model, a single A100 40G/80G or V100 32G GPU is sufficient. To optimize efficiency, we recommend dividing the inference tasks into smaller chunks and running them in parallel across both high-end GPUs like the A100 and lower-end GPUs like the V100.
>
> Finally, we sincerely appreciate your meaningful questions. We will update the details of our discussion in the later revision.

---

### Official Review · Reviewer_EPjK · 2024-11-04

**Soundness:** 4
**Presentation:** 4
**Contribution:** 4
**Rating:** 8
**Confidence:** 3

**Summary:**

The authors propose a technique to generate pseudo-feedback using test cases. For math problem-solving, the test cases are unique (since each math problem has a singular, numerical answer, and a model either produces or doesn’t produce that answer). For code generation, multiple test cases are produced: the paper explores both test case generation using a frontier LLM and using the model being trained.

This approach provides a new method that does not rely on the high-quality human-verified data required for DPO. Implementing this approach with Llama-3.1-8B, Mathstral-7B, and Deepseek-coder-7B-v1.5-Instruct for math and code reasoning respectively, shows improvements from 5 to over 10% across various benchmarks.

**Strengths:**

* Creates the pseudo feedback framework in a way that combines both math and code reasoning tasks.
* Because the pipeline for generating pseudo feedback is entirely automated, the process is scalable and less expensive than recreating such a process with humans would be.
* The experiments are robust, conducted with three open LLMs, and with multiple combinations of techniques across two domains.
* The results of the experiments are not only significant, but also well-explained. The authors do a good job of properly explaining any particularly interesting/surprising results.
* The approach using frontier LLMs removes the need for ground-truth labels, enabling better use of unlabeled reasoning data for model improvement.
* The authors provide an appendix with more interesting insights and all the necessary details needed to recreate their work/expand on it, making for a meaningful contribution to the field.

**Weaknesses:**

* The primary weaknesses I see with this approach are the limitations caused by self-consistency over self-generated pseudo feedback. I could see improvement over iterations being marginal for more challenging math/coding problems, as the self-consistent "answer" for them is likely incorrect.

**Questions:**

* Same as the weakness — I’d love to see whether performance on a subset of more challenging problems (which could be measured by looking at problems where the base models provided an incorrect answer even after applying self-consistency) improved over time with self-generated pseudo-feedback.

---

> ### Author Response · Authors · 2024-11-22
> **Response - 1 / 3**
>
> We sincerely appreciate it for taking the time to review our submission and providing valuable feedback.
>
> >I could see improvement over iterations being marginal for more challenging math/coding problems, as the self-consistent "answer" for them is likely incorrect.
>
> ### **Correctness of self-consistent "answers" (or test-cases)**
> As you recommended, we first examined the correctness of “self-consistent answers (or test cases)". We choose to conduct our experiments on coding, since APPs is part of our training dataset with gold program solution annotation (see line 300), which can be used to verify our synthetic test cases for each iteration.
>
> Specifically, for each question in the APPs training set, we selected a ground-truth program to evaluate the synthetic test cases. A synthetic test case is valid, if and only if given the test case input and its output matches the output of the ground-truth program.
> We calculate the "Test Case Pass Rate," which represents the percentage of valid test cases among the total generated.
> The results are as follows, where the `Model` column denotes the model optimized by the pseudo feedback constructed from the corresponding test cases.
>
> | Model                           | Test Case Pass Rate |
> |---------------------------------|:---------------------:|
> | PFPO-Self Iter. 0 (APPs)        | 62.68              |
> | PFPO-Self Iter. 1 (APPs & M.C.) | **66.80**              |
> | PFPO-Self Iter. 2 (APPs & M.C. & xCode.) | 66.75       |
>
>
>
> Indeed, the quality of synthetic test cases of Iter. 1 is the best and Iter. 2 is slightly worse and the final coding performance follows a similar trend (see Table 3). We assume that the quality of synthetic test cases can be one of the contributing factors to the marginal improvement over iterations. We will add these analyses to our updated manuscript.

---

> > ### Comment · Reviewer_EPjK · 2024-11-25
> > **Thank you!**
> >
> > Thanks for responding to all my concerns! I believe the addition of analyses on the challenging problems should be beneficial to supporting the claims made in the paper. I strongly support the acceptance of this paper!

---

> ### Author Response · Authors · 2024-11-22
> **Response - 2 / 3**
>
> ### **Reasons and Potential solutions for improvement over iterations being marginal**
>
> As also observed in [1][2], iterative style DPO tends to saturate after several iterations, regardless of whether the preference pairs are human labeled or synthetic. We believe there are at least two reasons for the saturation. The first one (as mentioned earlier) is the quality of synthetic test cases (or answers) may plateau at some point.
>
> The second reason could be that as the model improves with each iteration, more problems in the training set become "too easy" for the model at its current stage. For extremely challenging problems where no correct solution can be generated despite many attempts (sampling), we are unable to create valid preference pairs and are thus unlikely to solve them in the next iteration. The number of moderately challenging problems, which are most effective for improving the model, gradually decreases after each iteration.
>
> **Potential Solutions:** One possible solution to this is introducing unseen problems in each iteration to increase the likelihood of hitting new moderately challenging problems.
>
> In the third block of Table 1 (line 336 to 342), we simulated the above setting by introducing new problems across iterations. As detailed in Appendix A, we split the NuminaMath dataset into five subsets of equal size and use a different subset for each iteration. We observed consistent improvements across iterations (see w/ PFPO-Self Iter. 1 to 5 rows in Table 1).
>
> However, please note that applying the above method requires a significant amount of unseen high-quality questions in that domain. Therefore, we cannot apply this strategy with “Mathstral-7B-v0.1” based models (line 343 to 347; Mathstral-7B-v0.1 has been trained on Numina already) or in coding domain (high quality prompts in coding domain is very limited).
>
> Finally, it is worth noting that our method’s advantage lies in its scalability, as no additional human annotation for answers is required when incorporating new problems.
>
> [1] Iterative Preference Learning from Human Feedback: Bridging Theory and Practice for RLHF under KL-constraint
> [2] Bootstrapping Language Models with DPO Implicit Rewards
>
> Thanks again for your insightful question. All results and discussions above will be added to our updated manuscript.

---

> ### Author Response · Authors · 2024-11-22
> **Response - 3 / 3**
>
> > Same as the weakness — I’d love to see whether performance on a subset of more challenging problems (which could be measured by looking at problems where the base models provided an incorrect answer even after applying self-consistency) improved over time with self-generated pseudo-feedback.
>
> Thank you for your valuable suggestion. We evaluated our performance on a subset of challenging math problems. We created the challenging math test set as follows.
> We began by sampling 16 solutions generated by Mathstral-7B w/ SFT on the MATH test set. We then identified questions for which the predicted answers, even after applying majority voting, remained incorrect. This yielded a MATH-Challenging test set of 1,580 questions. The results of our models using FPPO on this test set are as follows:
>
> | Mathstral-7B-v0.1          | Greedy Decoding |
> |----------------------------|:-----------------:|
> | w/ SFT                    | 19.6            |
> | w/ PFPO-LLM-Iter 0        | 23.6            |
> | w/ PFPO-Self-Iter 1       | 24.6            |
> | w/ PFPO-Self-Iter 2       | **26.7**            |
>
> The results indicate that our method can achieve improvements even on challenging questions over iterations.
>
> All results and discussions above will be added to our updated manuscript.

---

### Meta-Review · Area_Chair_Y672 · 2024-12-23

**Metareview:**

This paper aims to alleviate the problem of scarcity of preference data, and propose to methods to generate pseudo-feedback in the context of reasoning problems such as math and coding by using the evaluation on test cases as the feedback.  The paper explores two forms of pseudo feedback, and convincingly demonstrate the effectiveness of the proposed methods on benchmark reasoning datasets.

While there were slight concern on the technical novelty of the proposed methods, all reviewers agree the empirical evidence is convincing on the effectiveness of the proposed methods in practice. We thus recommend acceptance.

**Additional Comments On Reviewer Discussion:**

All reviewers reach consensus recommending the acceptance of this paper after rebuttal.

---

### Decision · Program_Chairs · 2025-01-22

Accept (Spotlight)